# Native KCC2 interactome reveals PACSIN1 as a critical regulator of synaptic inhibition

**Vivek Mahadevan[1‡], C Sahara Khademullah[1†], Zahra Dargaei[1†], Jonah Chevrier[1], Pavel Uvarov[2], Julian Kwan[3], Richard D Bagshaw[4], Tony Pawson[4§], Andrew Emili[3], Yves De Koninck[5,6], Victor Anggono[7], Matti Airaksinen[2], Melanie A Woodin[1]\***

[1]Department of Cell and Systems Biology, University of Toronto, Toronto, Canada; [2]Department of Anatomy, Faculty of Medicine, University of Helsinki, Helsinki, Finland; [3]Department of Molecular Genetics, Donnelly Centre for Cellular and Biomolecular Research, University of Toronto, Toronto, Canada; [4]Lunenfeld-Tanenbaum Research Institute, Mount Sinai Hospital, Toronto, Canada; [5]Institut Universitaire en Santé Mentale de Québec, Québec, Canada; [6]Department of Psychiatry and Neuroscience, Université Laval, Québec, Canada; [7]Queensland Brain Institute, Clem Jones Centre for Ageing Dementia Research, The University of Queensland, Brisbane, Australia

**\*For correspondence:**
m.woodin@utoronto.ca

[†]These authors contributed equally to this work

**Present address:** [‡]Section on Cellular and Synaptic Physiology, Eunice Kennedy Shriver National Institute of Child Health and Human Development, Bethesda, United States

[§]Deceased

**Competing interests:** The authors declare that no competing interests exist.

**Abstract** KCC2 is a neuron-specific K$^+$-Cl$^-$ cotransporter essential for establishing the Cl$^-$ gradient required for hyperpolarizing inhibition in the central nervous system (CNS). KCC2 is highly localized to excitatory synapses where it regulates spine morphogenesis and AMPA receptor confinement. Aberrant KCC2 function contributes to human neurological disorders including epilepsy and neuropathic pain. Using functional proteomics, we identified the KCC2-interactome in the mouse brain to determine KCC2-protein interactions that regulate KCC2 function. Our analysis revealed that KCC2 interacts with diverse proteins, and its most predominant interactors play important roles in postsynaptic receptor recycling. The most abundant KCC2 interactor is a neuronal endocytic regulatory protein termed PACSIN1 (SYNDAPIN1). We verified the PACSIN1-KCC2 interaction biochemically and demonstrated that shRNA knockdown of PACSIN1 in hippocampal neurons increases KCC2 expression and hyperpolarizes the reversal potential for Cl$^-$. Overall, our global native-KCC2 interactome and subsequent characterization revealed PACSIN1 as a novel and potent negative regulator of KCC2.
DOI: https://doi.org/10.7554/eLife.28270.001

## Introduction

GABA and glycine are the key inhibitory neurotransmitters of the mature nervous system, and most synaptic inhibition is mediated by Cl$^-$ permeable GABA$_A$ and glycine receptors. This hyperpolarizing inhibition results from the inward gradient for Cl$^-$ established primarily by the K$^+$-Cl$^-$ cotransporter KCC2, which exports Cl$^-$ to maintain low intracellular Cl$^-$ [Cl$^-$]$_i$ (*Rivera et al., 1999*; *Doyon et al., 2016*). KCC2 is a member of the nine-member family of cation-chloride cotransporters and is unique among the members because it is present exclusively in neurons of the CNS, and mediates the electroneutral outward cotransport of K$^+$ and Cl$^-$. KCC2 protein is encoded by the *SLC12A5* gene, which via alternative splicing results in two transcript variants encoding the isoforms KCC2a and KCC2b (*Payne et al., 1996*; *Uvarov et al., 2007*).

**eLife digest** Neurons in the brain talk to each other by releasing chemicals called neurotransmitters. These neurotransmitters can either increase ('excite') or decrease ('inhibit') the activity of other neurons. Inhibitory neurotransmission uses the chemical GABA as a neurotransmitter. When a neuron releases GABA it is like applying the brake in your car – you can slow down subtly to stay under the speed limit, or stomp on it to avoid an accident. The brain needs to carefully control the amount of inhibition so that the animal can learn and produce complex behaviours.

For GABA to inhibit the activity of a neuron, the neuron must maintain a low amount of chloride ions inside. A transporter protein called KCC2 shuttles chloride out of cells; if this transporter fails to work, chloride builds up in the neuron and prevents inhibition so that GABA neurotransmission switches from inhibitory to excitatory. This breakdown of GABA inhibition is a hallmark of abnormal brain activity during conditions such as epilepsy, pain and some forms of autism.

Despite the fact that neurons need KCC2 for inhibition in the brain, we do not know much about how this transporter works. Since the activity of a protein is determined in part by the other proteins it interacts with, it is therefore important to identify all the proteins that interact with KCC2 – termed the KCC2 interactome.

To discover these protein interactions, Mahadevan et al. performed a technique called liquid chromatography-mass spectrometry on KCC2 protein isolated from mouse brains. This revealed that there are 181 proteins in the KCC2 interactome. Of these proteins, the most abundant was a protein called PACSIN1, which helps to pull proteins out of the membrane that surrounds each neuron.

To investigate how the interaction between PACSIN1 and KCC2 regulates the activity of this transporter, Mahadevan et al. performed fluorescence imaging of neurons and recorded their electrical activity. This revealed that PACSIN1 restricts the expression of KCC2, meaning that the more PACSIN there is in the neuron, the less KCC2 will be present.

The KCC2 interactome provides a database of proteins that can be targeted to increase the activity of KCC2. This could allow new treatments to be developed for brain disorders in which the inhibition of neurons is reduced.

DOI: https://doi.org/10.7554/eLife.28270.002

During embryonic development, KCC2 expression is low and GABA and glycine act as excitatory neurotransmitters; however, during early postnatal development KCC2 expression is dramatically upregulated and GABA and glycine become inhibitory (*Ben-Ari, 2002*; *Blaesse et al., 2009*). Excitation-inhibition imbalance underlies numerous neurological disorders (*Kahle et al., 2008*; *Nelson and Valakh, 2015*), and in many of these disorders, the decrease in inhibition results from a reduction in KCC2 expression. In particular, KCC2 dysfunction contributes to the onset of seizures (*Huberfeld et al., 2007*; *Kahle et al., 2014*; *Puskarjov et al., 2014*; *Stödberg et al., 2015*; *Saitsu et al., 2016*), neuropathic pain (*Coull et al., 2003*), schizophrenia (*Tao et al., 2012*), and autism spectrum disorders (ASD) (*Cellot and Cherubini, 2014*; *Tang et al., 2016a*; *Banerjee et al., 2016*). Despite the critical importance of this transporter in maintaining inhibition and proper brain function, our understanding of KCC2 regulation is rudimentary.

In addition to its canonical role of $Cl^-$ extrusion that regulates synaptic inhibition, KCC2 has also emerged as a key regulator of excitatory synaptic transmission. KCC2 is highly localized in the vicinity of excitatory synapses (*Gulyás et al., 2001*; *Chamma et al., 2013*) and regulates both the development of dendritic spine morphology (*Li et al., 2007*; *Chevy et al., 2015*; *Llano et al., 2015*) and function of AMPA-mediated glutamatergic synapses (*Gauvain et al., 2011*; *Chevy et al., 2015*; *Llano et al., 2015*). Thus, a dysregulation of these non-canonical KCC2 functions at excitatory synapses may also contribute to the onset of neurological disorders associated with excitation-inhibition imbalances.

KCC2 is regulated by multiple posttranslational mechanisms including phosphoregulation by distinct kinases and phosphatases (*Lee et al., 2007*; *Kahle et al., 2013*; *Medina et al., 2014*), lipid rafts and oligomerization (*Blaesse et al., 2006*; *Watanabe et al., 2009*), and protease-dependent cleavage (*Puskarjov et al., 2012*). KCC2 expression and function is also regulated by protein interactions,

including creatine kinase B (CKB) (*Inoue et al., 2006*), sodium/potassium ATPase subunit 2 (ATP1A2) (*Ikeda et al., 2004*), chloride cotransporter interacting protein 1 (CIP1) (*Wenz et al., 2009*), protein associated with Myc (PAM) (*Garbarini and Delpire, 2008*), 4.1N (*Li et al., 2007*), the glutamate receptor subunit GluK2, its auxiliary subunit Neto2 (*Ivakine et al., 2013*; *Mahadevan et al., 2014*; *Pressey et al., 2017*), cofilin1 (CFL1) (*Chevy et al., 2015*; *Llano et al., 2015*), the GABA$_B$ receptor subunit GABA$_B$R1 (*Wright et al., 2017*), metabotropic glutamate receptor subunits mGluR1/5 (*Farr et al., 2004*; *Banke and Gegelashvili, 2008*; *Mahadevan and Woodin, 2016*; *Notartomaso et al., 2017*), and RAB11(*Roussa et al., 2016*). However, since KCC2 exists in a large multi-protein complex (MPC) (*Mahadevan et al., 2015*), it is likely that these previously identified interactions do not represent all of the components of native-KCC2 MPCs.

In the present study, we performed unbiased affinity purifications (AP) of native-KCC2 coupled with high-resolution mass spectrometry (MS) using three different KCC2 epitopes from whole-brain membrane fractions prepared from developing and mature mouse brain. We found that native KCC2 exists in macromolecular complexes comprised of interacting partners from diverse classes of transmembrane and soluble proteins. Subsequent network analysis revealed numerous previously unknown native-KCC2 protein interactors related to receptor recycling and vesicular endocytosis functions. We characterized the highest-confidence KCC2 partner identified in this screen, PACSIN1, and determined that PACSIN1 is a novel and potent negative regulator of KCC2 expression and function.

## Results

### Determining affinity purification (AP) conditions to extract native-KCC2

In order to determine the composition of native KCC2 MPCs using AP-MS, we first determined the detergent-based conditions that preserve native KCC2 following membrane extraction. In a non-denaturing Blue-Native PAGE (BN-PAGE), native-KCC2 migrated between 400 kDa – 1000 kDa in the presence of the native detergents $C_{12}E_9$ (nonaethylene glycol monododecyl ether), CHAPS (3-[(3-Cholamidopropyl) dimethylammonio]−1-propanesulfonate hydrate) 1, and DDM (n-dodecyl β-d-maltoside). However, all other detergent compositions previously used for KCC2 solubilization resulted in KCC2 migration at lower molecular weights (*Figure 1a*). This indicates that native detergent extractions are efficient at preserving higher order KCC2 MPCs. Upon further analysis using standard SDS-PAGE, we observed that the total KCC2 extracted was greater in $C_{12}E_9$ and CHAPS-based detergent extractions in comparison with all other detergents (*Figure 1a*, *Figure 1—figure supplements 1* and *2*), hence we restricted our further analysis to $C_{12}E_9$ and CHAPS-based membrane preparations.

To determine which of these two detergents was optimal for our subsequent full-scale proteomic analysis, we performed AP-MS to compare the efficacy of $C_{12}E_9$ versus CHAPS-solubilized membrane fractions. Immunopurification was performed on membrane fractions prepared from adult (P50) wild-type (WT) mouse brain, using a well-validated commercially available C-terminal KCC2 antibody (*Williams et al., 1999*; *Gulyás et al., 2001*; *Woo et al., 2002*; *Mahadevan et al., 2014*) and a control IgG antibody. In both detergent conditions, we recovered maximum peptides corresponding to KCC2 from the KCC2 pull downs (KCC2-AP), in comparison to the control IgG pull downs (IgG-AP), confirming the specificity of the C-terminal KCC2 antibody (*Figure 1b* and *Figure 1—source data 1*). However, upon further examination, two key pieces of evidence indicated that $C_{12}E_9$ -based conditions are optimal for proteomic analysis of native KCC2. First, we observed peptides corresponding to both KCC2 isoforms-a and -b; in $C_{12}E_9$-based samples, but we could only detect peptides corresponding to KCC2 isoform-a in KCC2-APs from CHAPS-based samples. Second, we observed a higher enrichment of peptides corresponding to previously identified KCC2 interactors belonging to the family of $Na^+/K^+$ ATPases (ATP1A1-3), and the family of creatine kinases (CKB, CKMT1), and cofilin1 (CFL1) in the KCC2-AP from $C_{12}E_9$-based samples. Based on these results, we concluded that $C_{12}E_9$-based solubilization conditions yield more KCC2-specific binding partners and fewer IgG-specific binding partners compared to CHAPS, and thus provide a higher stringency for KCC2 AP-MS. Thus, we performed all subsequent proteomic analysis of native KCC2 on samples solubilized with $C_{12}E_9$.

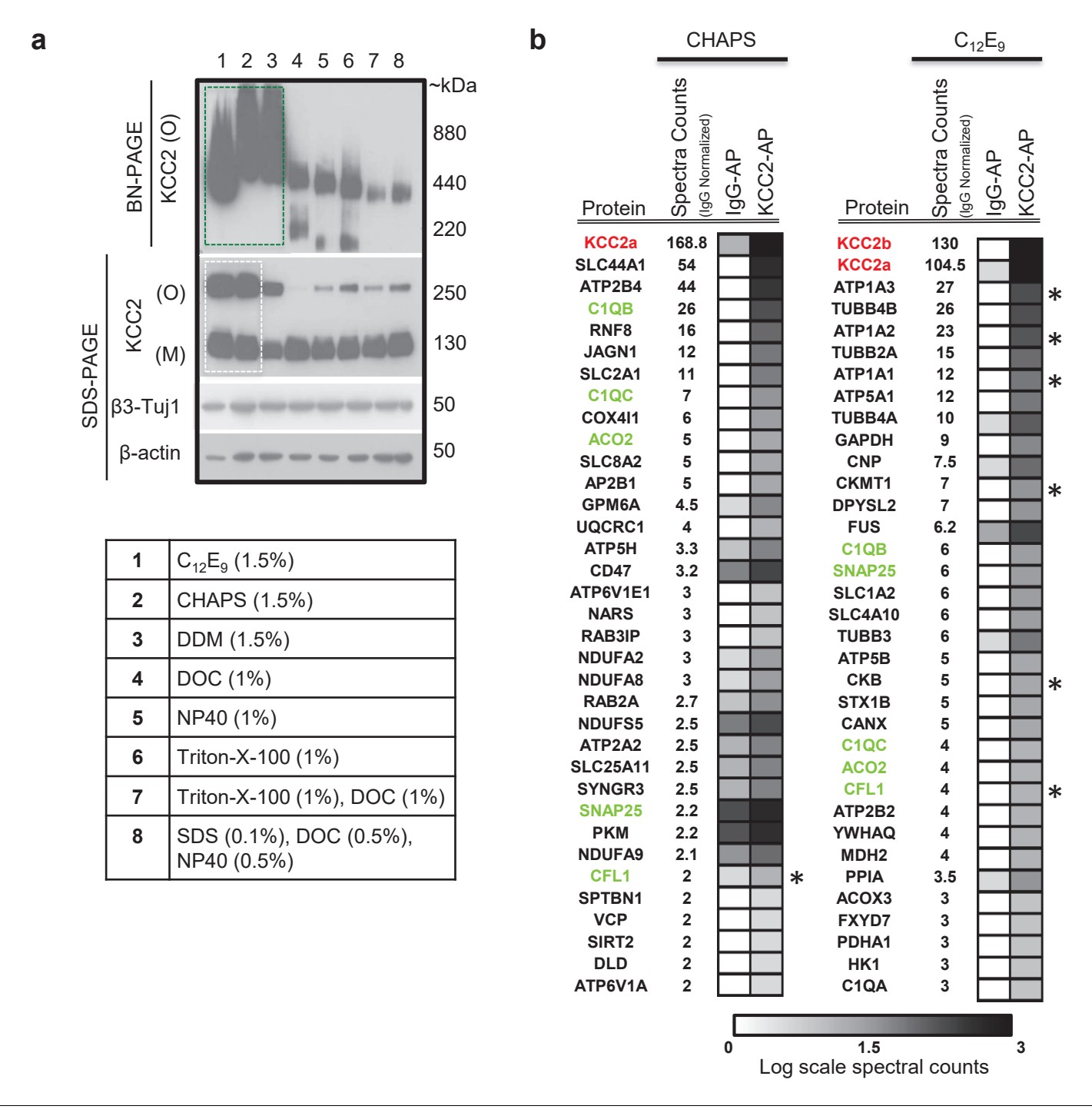

**Figure 1.** KCC2 multi-protein complexes can be extracted using native detergents. (**a**) BN-PAGE and SDS-PAGE separation of solubilized membrane fractions prepared from ~P50 mouse brain, using the detergents listed in the associated table. Protein separations were western-blotted and probed with antibodies indicated on the left. O, oligomer; M, monomer. Blots are representative of two independent biological replicates. (**b**) Comparison of the top 35 proteins identified with high confidence in C-terminal KCC2 antibody immunoprecipitations from CHAPS-based or $C_{12}E_9$-based membrane extractions. IgG-AP immunoprecipitations were performed as a control. Heat maps represent log scale spectral counts of individual proteins per condition, expressed relative to global spectral counts. Unique peptides corresponding to KCC2 (indicated in red font) were most abundant in both conditions, confirming the specificity of the C-terminal antibody. Previously identified KCC2 interacting partners are identified by asterisks. Proteins in green represent those that commonly co-precipitated with KCC2 regardless of the detergent extraction.

DOI: https://doi.org/10.7554/eLife.28270.003

*Figure 1 continued on next page*

*Figure 1 continued*

The following source data and figure supplements are available for figure 1:

**Source data 1.** Proteins enriched in KCC2-AP using CHAPS vs C12E9.
DOI: https://doi.org/10.7554/eLife.28270.006
**Figure supplement 1.** Workflow to enrich KCC2 complexes.
DOI: https://doi.org/10.7554/eLife.28270.004
**Figure supplement 2.** SDS-PAGE separation of solubilized membrane fractions.
DOI: https://doi.org/10.7554/eLife.28270.005

## Multi-epitope (ME) proteomic analysis of KCC2 complexes in the developing and mature brain

To focus our proteomic analysis on KCC2b, which is the abundant isoform primarily responsible for the shift from excitatory to inhibitory GABA during early postnatal development (*Uvarov et al., 2007*; *Kaila et al., 2014*), we used a multi-epitope approach that allowed us to distinguish the KCC2 isoforms (*Figure 2a*). The C-terminal antibody recognizes both isoforms (*Uvarov et al., 2007*; *2009*), so we chose to use another antibody that is specifically raised against the unique N-terminal tail of the KCC2b isoform. Lastly, we used a phosphospecific antibody for serine 940 (pS940), as phosphorylation of this residue increases KCC2 surface expression and/or transporter function (*Lee et al., 2007*; *Lee et al., 2011*). We validated these three KCC2 antibodies (C-terminal, N-terminal, and pS940) for KCC2-immunoenrichment (*Figure 2—figure supplement 1*) (*Mahadevan et al., 2014*). Moreover, by taking a multi-epitope approach, we significantly increased the likelihood of detecting KCC2 interactions that may be missed during single-epitope AP approaches. We performed 34 rounds of AP-MS including both developing (P5) and mature/adult (P50) WT mouse brain lysates (*Figure 2—source data 1*). We could not use KCC2-knockout brains since these animals die at birth, so as an alternative we used a mock IP for each sample condition in the absence of the KCC2 antibody using parallel preimmunization immunoglobulin (IgG/IgY) as negative controls. We obtained 440 potential KCC2 protein interactors with 99% confidence and a 1% false discovery rate. We identified KCC2 peptides spanning the entire sequence of KCC2 with ~44% sequence coverage, primarily at the C- and N-terminal tails (*Figure 2b*); and in both the developing and mature brain, KCC2 peptides were the most abundant peptides identified in the KCC2-IPs (*Figure 2c*). While the KCC2 C-terminal antibody recovered peptides from both isoforms of KCC2, the N-terminal KCC2b-specific antibody did not recover any KCC2a isoform-specific peptides, indicating the specificity of the antibodies used, and the success of KCC2-immunoenrichment in our AP-assays.

## The KCC2 interactome

To build the KCC2 interactome, all potential KCC2 protein interactors were filtered according to their spectral count enrichment in the KCC2-APs, and normalized to IgG IPs. In the first pass filter grouping, we included proteins with at least two unique peptides and peptide-spectrum matches and a 3-fold increase in KCC2 spectral counts in the KCC2-AP in comparison to IgG-AP (*Figure 3—source data 1*). This yielded ~75 putative-KCC2 partners. In the second pass filter grouping, we identified additional putative-KCC2 partners by including those with only one unique peptide, or less than three-fold KCC2-AP enrichment, if they met one of the following criteria: (a) the protein was a previously validated KCC2 physical/functional interactor; (b) the protein family already appeared in the first-pass filter; (c) the protein appeared as a single-peptide interactor across multiple experiments (e.g. multiple antibodies, or in lysates from both age timepoints, or in both replicates). Including these additional proteins from the second pass filtering yielded 186 putative-KCC2 partners. We next eliminated the 36 proteins that have been previously identified as commonly occurring spurious interactors in LC/MS experiments as indicated in the CRAPome database (*Figure 3—source data 2*) (*Mellacheruvu et al., 2013*). By applying these filtering criteria and processes, we established a total list of 150 putative KCC2 partners in our present LC-MS assay (*Figure 3—source data 3*). More than half of these KCC2 interactors were exclusively enriched in KCC2-APs from the mature brain (85 proteins,~57% overlap), while approximately one-third (41 proteins,~27% overlap) were identified across both the developing and mature brain (*Figure 3—*

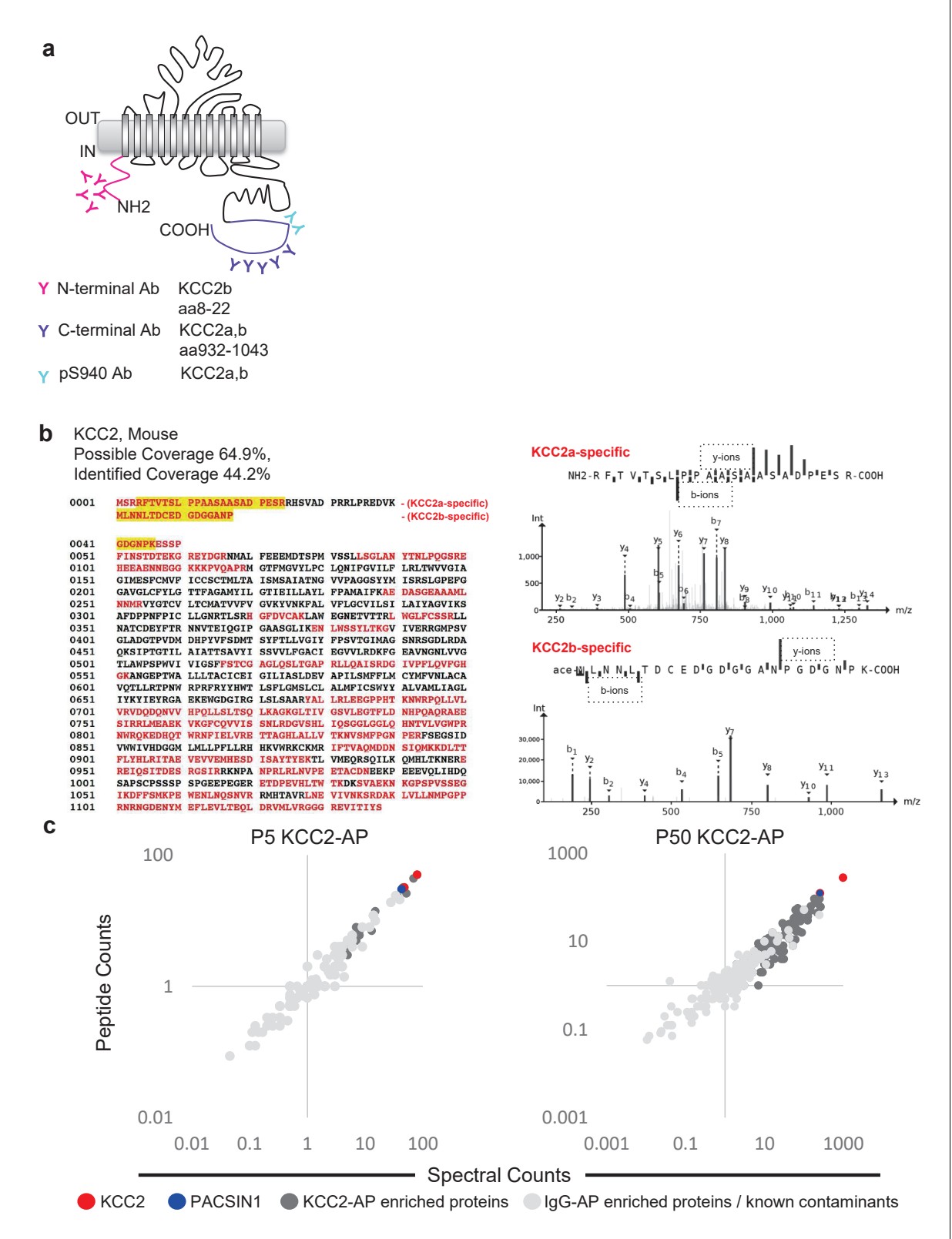

**Figure 2.** Multi-epitope AP identifies native-KCC2 protein constituents in mouse brain. (**a**) Schematic of the locations of anti-KCC2 antibodies. (**b**) The primary KCC2 amino acid sequence. Red indicates the protein coverage of KCC2 identified by MS analysis; yellow indicates unique coverage for KCC2a and KCC2b isoforms. MS/MS- spectra of peptides unique for KCC2a and KCC2b. Right: the MS/MS ion fragmentation of the corresponding amino acid sequence is indicated above the spectra. (**c**) Spectral and peptide count plots of proteins in AP with all three anti-KCC2 antibodies in developing brain

*Figure 2 continued on next page*

*Figure 2 continued*

membrane fractions (P5, left) and adult brain membrane fractions (P50, right). Peptide and spectral counts are normalized (anti-KCC2/IgG) and plotted on a log scale. Red circles - highly enriched KCC2 bait. Blue circles - highly enriched PACSIN1 target peptides. Dark-grey circles - top proteins enriched with KCC2-AP in comparison to IgG control-AP. Light-grey circles - proteins enriched in IgG control-AP in comparison to KCC2-AP and known spurious interactors.

DOI: https://doi.org/10.7554/eLife.28270.007

The following source data and figure supplement are available for figure 2:

**Source data 1.** LC/MS replicates.
DOI: https://doi.org/10.7554/eLife.28270.009
**Figure supplement 1.** Validation of KCC2 antibodies for immunodepletion.
DOI: https://doi.org/10.7554/eLife.28270.008

*figure supplement 1*). Only relatively small percentages were exclusively enriched in the developing brain (24 proteins,~16% overlap).

Next, we segregated the 150 putative KCC2 partners into high-confidence (platinum, gold), moderate-confidence (silver), or lower confidence (bronze) putative KCC2-interactors (*Figure 3*, *Table 1* and *Figure 3—source data 4*). This segregation was based on the largest probability of a bait-prey pair across all replicate purifications, as indicated by the MaxP score (*Choi et al., 2012*), and the presence of a particular protein across replicates, and spectral enrichment of a particular protein across all experiments. Platinum KCC2-partners are those proteins enriched in a minimum of two out of three replicates, show 5 + fold spectral enrichment, and a MaxP SAINT score ≥0.89. Gold KCC2-partners were those proteins enriched in only one replicate with normalized spectral count enrichments ≥ 5 and a MaxP SAINT score ≥0.89. Silver KCC2-partners were those with normalized spectral count enrichments between 3 and 5, and a MaxP score between 0.7 and 0.89. Bronze KCC2-parterns were all remaining proteins that were not designated as Platinum, Gold or Silver. Lastly, we added 31 proteins that have been previously established as KCC2-physical/functional partners but were not identified in our present LC-MS assay (*Figure 3—source data 4*). The proteins identified in the present screen (150) and the proteins previously established as key KCC2 physical/functional partners (31), together constitutes the 181 members of the proposed KCC2 interactome. All 181 proteins were included in subsequent network analyses.

## Members of KCC2 interactome are highly represented at excitatory synapses

To interpret the potential functional role of KCC2-protein interactors, we first segregated them based on their abundance at excitatory and inhibitory synapses. To perform this analysis, we mapped the KCC2 interactome to the excitatory synapse-enriched postsynaptic density (PSD), Nlgn1, Lrrtm1, and Lrrtm2 proteomes (*Collins et al., 2006*; *Loh et al., 2016*), or the inhibitory synapse-enriched iPSD, GABA$_A$R, GABA$_B$R, NLGN2, Slitrk3, and GlyR proteomes (*Heller et al., 2012*; *Del Pino et al., 2014*; *Kang et al., 2014*; *Loh et al., 2016*; *Nakamura et al., 2016*; *Schwenk et al., 2016*; *Uezu et al., 2016*) (*Figure 4—source data 1*). Interactome mapping revealed that ~43% of proteins in the KCC2 interactome (77/181) were exclusively enriched at excitatory synapses, while only ~2% of proteins (4/181) were exclusively enriched at inhibitory synapses (*Figure 4a,b*). However, ~15% proteins (30/181) were mapped to both excitatory and inhibitory synapses, whereas ~39% proteins (70/181) did not map to either synapses.

To further examine the KCC2 interactome based on cellular functions, we performed an Ingenuity Pathway Analysis (IPA) to segregate the KCC2-interactors into highly enriched Gene Ontology (GO) classes. Performing this IPA analysis revealed that KCC2 partners segregate into multiple cellular and molecular functional nodes, which we then combined into three broad categories that collectively had high p values: ion homeostasis, dendritic cytoskeleton rearrangement, and receptor trafficking (*Figure 4c–e*; *Figure 4—source data 2*). KCC2 has been previously associated with both ion homeostasis and dendritic spine morphology, and consistent with this previous work we identified previously characterized KCC2 functional or physical interactors, including subunits of the sodium/potassium (Na$^+$/K$^+$) ATPase, including the previously characterized KCC2 interactor ATP1A2 (*Ikeda et al., 2004*), and Cofilin1, which was recently demonstrated to be important for KCC2-mediated plasticity at excitatory synapses (*Chevy et al., 2015*; *Llano et al., 2015*). The third category,

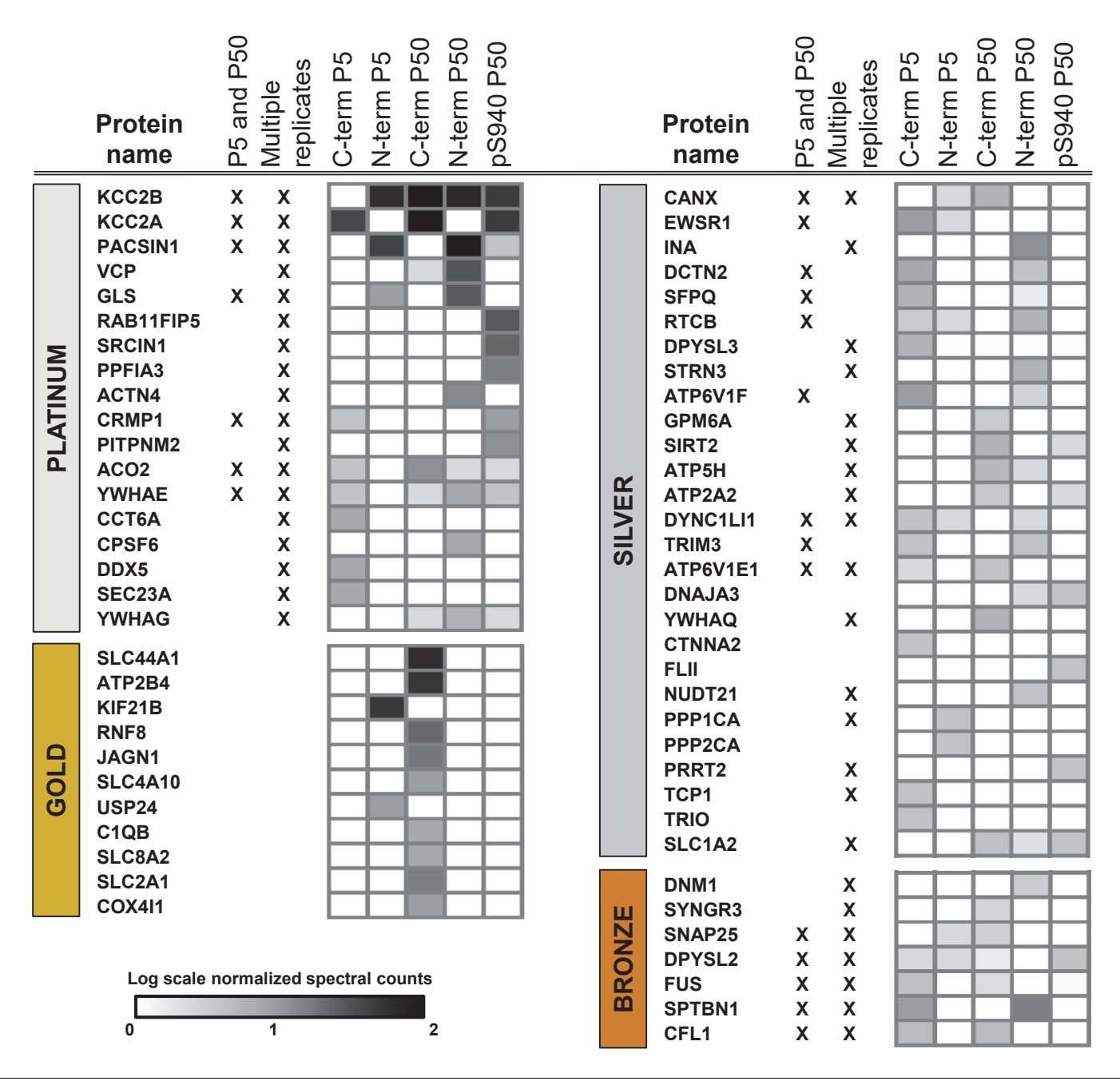

**Figure 3.** ME-AP reveals distinct KCC2 constituents in developing and mature brain. Summary of the top 70 proteins identified with high confidence across KCC2-ME AP in the developing and mature brain. PLATINUM interactors: proteins enriched in a minimum of 2/3 replicates, and show 5 + fold spectral enrichment. GOLD interactors: proteins with 5 + fold spectral enrichment in one replicate. SILVER interactors: 3–5 fold spectral enrichment; BRONZE interactors: 1.5–3 fold spectral enrichment. Enrichment is in KCC2-AP in comparison with IgG-AP. Heat map represents log scale spectral enrichment of individual proteins per antibody condition, relative to respective control conditions. See *Table 1* for a list of the transmembrane and soluble KCC2 interactors. .

DOI: https://doi.org/10.7554/eLife.28270.010

The following source data and figure supplements are available for figure 3:

**Source data 1.** All validated peptide-spectrum matches for KCC2-AP and IgG-AP.
DOI: https://doi.org/10.7554/eLife.28270.013

**Source data 2.** Top CRAPome members that appear at 50% frequency.
DOI: https://doi.org/10.7554/eLife.28270.014

*Figure 3 continued on next page*

*Figure 3 continued*

**Source data 3.** KCC2 interactors identified in AP/MS categorized into platinum, gold, silver, and bronze.
DOI: https://doi.org/10.7554/eLife.28270.015
**Source data 4.** Previously established KCC2 partners not identified in the present screen.
DOI: https://doi.org/10.7554/eLife.28270.016
**Figure supplement 1.** ME-AP proteomics identify the protein constituents of native KCC2.
DOI: https://doi.org/10.7554/eLife.28270.011
**Figure supplement 2.** Workflow for curating the KCC2 interactome.
DOI: https://doi.org/10.7554/eLife.28270.012

receptor trafficking, has a denser network (clustering coefficient of 0.63 and an average of ~4.4 neighbors) in comparison to the other networks, suggesting a tight link between KCC2 and proteins in this node. Notably, this analysis revealed multiple novel putative-KCC2 partners, including PACSIN1, SNAP25, RAB11FIP5, CSNK2A1, DNM1 and AP2B1. All these novel putative interactors have established functions in membrane recycling and/or trafficking of glutamate receptor subunits (*Carroll et al., 1999*; *Lee et al., 2002*; *Vandenberghe et al., 2005*; *Pérez-Otaño et al., 2006*; *Selak et al., 2009*; *Sanz-Clemente et al., 2010*; *Anggono et al., 2013*; *Bacaj et al., 2015*). In order to determine the spatiotemporal expression profiles of the KCC2 interactome, we first performed transcriptomic analysis and hierarchical clustering of high-resolution human brain RNAseq data (*Kang et al., 2011*; *Willsey et al., 2013*) (obtained from http://brain-map.org). We observed that *SLC12A5* mRNA is expressed with several members of the receptor trafficking node in the hippocampus (*Figure 5a*), amygdala, thalamus, cerebellum and cortex (*Figure 5—figure supplement 1*; *Figure 5—source data 1*). The RNAseq data does not distinguish between isoforms KCC2a and KCC2b, which equally represented in the neonatal brain, while KCC2b is the predominant isoform in the adult brain.

In order to independently validate the utility of the KCC2 interactome, we proceeded to biochemical and functional analysis. We focused this validation analysis on proteins in the receptor trafficking category for two reasons: (i) the most abundant putative-KCC2 partner, PACSIN1 (PKC and CSNK2A1 substrate in neurons; also called as Syndapin1) is present in the receptor trafficking node; and (ii) the tightest KCC2-subnetwork exists in receptor trafficking node, indicating a dense interconnectivity between these proteins.

## PACSIN1 is a novel native-KCC2 binding partner

To biochemically and functionally validate our KCC2 interactome, we chose to focus on the putative KCC2-PACSIN1 interaction. The rationale for this selection was based on the following: (1) PACSIN one is the most abundant KCC2 interactor in our LC-MS assay (next only to KCC2), with a high normalized spectral count ratio and a high MaxP score, and with extensive amino acid sequence coverage (*Figure 5—figure supplement 2*); (2) PASCIN1 is a substrate for PKC, and PKC is a key regulator of KCC2 (*Lee et al., 2007*); (3) PASCIN1 is a substrate for CSNK2A1, and our LC-MS assay revealed CSNK2A1 also as a putative KCC2-interactor; (4) PACSIN1 is abundant at both excitatory and inhibitory synapses (*Pérez-Otaño et al., 2006*; *Del Pino et al., 2014*); and (5) PACSIN1 was identified as an abundant KCC2 interactor using two antibodies (N-terminal and pS940) in multiple replicates.

To independently verify whether KCC2 associated with PACSIN1, we performed a co-immunoprecipitation assays from adult whole brain native membrane preparations. We found that PACSIN1 was immunoprecipitated with an anti-KCC2b antibody, but not with control IgY antibodies (*Figure 5b*). Using a previously well-validated PACSIN1 antibody (*Anggono et al., 2013*), we confirmed this interaction in the reverse direction, indicating the existence of a KCC2-complex with PACSIN1 in vivo (*Figure 5b*). Consistent with our ability to co-immunoprecipitate KCC2 and PACSIN1, we found that the expression profiles of KCC2 and PACSIN1 are temporally aligned in the mouse brain (*Figure 5c*). To determine whether native-KCC2 complexes are stably associated with PACSIN1, we performed an antibody-shift assay coupled with two-dimensional BN-PAGE (2D BN-PAGE), which is a strategy that has been used previously to examine the native assemblies of several transmembrane protein multimeric complexes (*Schwenk et al., 2010*, *2012*), including that of native-KCC2 (*Mahadevan et al., 2014*, *2015*). Using this approach, we first verified that the addition

**Table 1.** KCC2 protein partners identified by ME-APs

| Protein name | UniProt ID | Spectral ratio | MaxP | ES | IS | P5 | P50 | pS940 |
|---|---|---|---|---|---|---|---|---|
| KCC2b | Q91V14-2 | 349.0 | 1 | X | X | X | X | X |
| KCC2a | Q91V14-1 | 203.8 | 1 | X | X | X | X | X |
| PACSIN1 | Q61644 | 136.0 | 1 | X | X | X | X | X |
| SLC44A1 | Q6X893 | 54.0 | 1 | | | | X | |
| ATP2B4 | Q6Q477 | 44.0 | 1 | X | | | X | |
| KIF21B | Q9QXL1 | 41.0 | 1 | | | X | | |
| VCP | Q01853 | 24.0 | 1 | X | | | X | |
| GLS | D3Z7P3 | 23.5 | 1 | | | X | X | |
| RAB11FIP5 | Q8R361 | 22.0 | 1 | X | | | X | X |
| SRCIN1 | Q9QWI6 | 17.0 | 1 | X | | | X | X |
| RNF8 | Q8VC56 | 16.0 | 1 | | | | X | |
| JAGN1 | Q5XKN4 | 12.0 | 1 | | | | X | |
| PPFIA3 | P60469 | 11.0 | 1 | X | | | X | X |
| SLC2A1 | P17809 | 11.0 | 0.96 | | | | X | |
| YWHAE | P62259 | 10.0 | 0.99 | X | X | X | X | X |
| ACTN4 | P57780 | 9.0 | 1 | X | | | X | |
| CRMP1 | P97427 | 8.0 | 1 | X | | X | X | X |
| PITPNM2 | Q6ZPQ6 | 8.0 | 1 | | | | X | X |
| ACO2 | Q99KI0 | 7.0 | 1 | X | | X | X | X |
| COX4I1 | P19783 | 6.0 | 0.96 | | | | X | |
| SLC4A10 | Q5DTL9 | 6.0 | 1 | X | X | | X | |
| USP24 | B1AY13 | 6.0 | 1 | | | X | | |
| YWHAG | P61982 | 6.0 | 0.96 | X | X | | X | X |
| C1QB | P14106 | 5.0 | 0.89 | | | | X | X |
| CCT6A | P80317 | 5.0 | 0.99 | X | | X | | |
| CPSF6 | Q6NVF9 | 5.0 | 0.99 | | | | X | |
| DDX5 | Q61656 | 5.0 | 0.99 | X | | X | | |
| DYNC1LI1 | Q8R1Q8 | 5.0 | 0.78 | X | | X | X | |
| SEC23A | Q01405 | 5.0 | 0.99 | X | X | X | | |
| SLC8A2 | Q8K596 | 5.0 | 0.99 | X | | | X | |
| TRIM3 | Q9R1R2 | 5.0 | 0.78 | X | X | X | X | |
| CANX | P35564 | 4.5 | 1 | X | X | X | X | |
| ATP5H | Q9DCX2 | 4.0 | 0.81 | | | | X | |
| ATP6V1E1 | P50518 | 4.0 | 0.78 | X | | X | X | |
| DDX17 | Q501J6 | 4.0 | 0.96 | | | X | | |
| DNAJA3 | Q99M87 | 4.0 | 0.78 | X | | | X | X |
| DPYSL3 | Q62188 | 4.0 | 0.96 | | | X | | |
| EWSR1 | Q61545 | 4.0 | 1 | | | X | X | X |
| MAG | P20917 | 4.0 | 0.96 | X | X | | X | |
| MDH2 | P08249 | 4.0 | 0.96 | X | | | X | |
| RTCB | Q99LF4 | 4.0 | 0.98 | | | X | X | |
| SNRPA | Q62189 | 4.0 | 0.96 | | | X | | |
| STOML2 | Q99JB2 | 4.0 | 0.96 | X | | X | | |
| STRN3 | Q9ERG2 | 4.0 | 0.96 | | | | X | |

*Table 1 continued on next page*

*Table 1 continued*

| Protein name | UniProt ID | Spectral ratio | MaxP | ES | IS | P5 | P50 | pS940 |
|---|---|---|---|---|---|---|---|---|
| YWHAQ | P68254 | 4.0 | 0.78 | X | | | X | |
| INA | P46660 | 3.6 | 1 | X | | | X | |
| ATP6V1F | Q9D1K2 | 3.5 | 0.95 | | | X | X | |
| DCTN2 | Q99KJ8 | 3.5 | 1 | X | | X | X | |
| CD47 | Q61735 | 3.2 | 0.9 | | X | | X | |
| ACOX3 | Q9EPL9 | 3.0 | 0.78 | | | | X | |
| AP2B1 | Q9DBG3 | 3.0 | 1 | X | X | | X | |
| APBA1 | B2RUJ5 | 3.0 | 0.78 | | | X | | |
| ATP2A2 | O55143 | 3.0 | 0.81 | X | | | X | X |
| CMPK1 | Q9DBP5 | 3.0 | 0.78 | X | | X | | |
| CRTC1 | Q68ED7 | 3.0 | 0.78 | | | | X | |
| CTNNA2 | Q61301 | 3.0 | 0.78 | X | | X | | |
| FLII | Q9JJ28 | 3.0 | 0.78 | | | | X | X |
| FXYD7 | P59648 | 3.0 | 0.78 | | | | X | |
| GPM6A | P35802 | 3.0 | 0.93 | X | | | X | X |
| GRID2 | Q61625 | 3.0 | 0.78 | X | | | X | |
| NARS | Q8BP47 | 3.0 | 0.78 | | | | X | |
| NDUFA2 | Q9CQ75 | 3.0 | 1 | | | | X | |
| NUDT21 | Q9CQF3 | 3.0 | 0.78 | | | | X | |
| PPP1CA | P62137 | 3.0 | 0.78 | X | X | X | | |
| PPP2CA | P63330 | 3.0 | 0.78 | X | | X | | |
| PRRT2 | E9PUL5 | 3.0 | 0.78 | | X | | X | X |
| SFPQ | Q8VIJ6 | 3.0 | 1 | X | | X | X | |
| SIRT2 | Q8VDQ8 | 3.0 | 0.89 | | | | X | X |
| SLC1A2 | P43006 | 3.0 | 0.69 | X | | | X | X |
| TCP1 | P11983 | 3.0 | 0.78 | X | | X | | |
| TRIO | Q0KL02 | 3.0 | 0.78 | X | | X | | |
| PTN | P63089 | 2.7 | 1 | | | X | | |
| RAB2A | P53994 | 2.7 | 0.84 | X | | | X | |
| NDUFS5 | Q99LY9 | 2.5 | 0.82 | | | | X | |
| CCT2 | P80314 | 2.5 | 1 | X | | X | | |
| DNM1 | P39053 | 2.5 | 1 | X | | | X | |
| SLC25A11 | Q9CR62 | 2.5 | 0.93 | X | X | | X | |
| DDX1 | Q91VR5 | 2.4 | 0.89 | X | | X | X | |
| NEDD4L | Q8CFI0 | 2.4 | 0.91 | X | X | X | X | |
| SYNGR3 | Q8R191 | 2.4 | 1 | X | | | X | X |
| DLD* | O08749 | 2.3 | 0.44 | X | | X | X | X |
| SNAP25 | P60879 | 2.3 | 0.65 | X | | X | X | X |
| DDX3X | Q62167 | 2.3 | 0.79 | X | | X | X | X |
| CAMK2G | Q923T9 | 2.3 | 1 | X | | | X | |
| FASN | P19096 | 2.3 | 1 | X | X | X | | |
| PKM | P52480 | 2.2 | 0.85 | | | | X | |
| NDUFA9 | Q9DC69 | 2.1 | 0.96 | X | | | X | |
| BASP1 | Q91XV3 | 2.1 | 0.72 | X | | | X | X |

*Table 1 continued on next page*

*Table 1 continued*

| Protein name | UniProt ID | Spectral ratio | MaxP | ES | IS | P5 | P50 | pS940 |
|---|---|---|---|---|---|---|---|---|
| CKB | Q04447 | 2.0 | 0.64 | X | | | X | |
| COX6C | Q9CPQ1 | 2.0 | 0.89 | | | | X | |
| CSNK2A1 | Q60737 | 2.0 | 0.84 | X | | X | X | |
| DHX9* | O70133 | 2.0 | 0.44 | | | X | X | |
| DPYSL2 | O08553 | 2.0 | 0.97 | X | | X | X | X |
| EDC4 | Q3UJB9 | 2.0 | 1 | | | | X | |
| FUS | P56959 | 2.0 | 0.94 | X | | X | X | X |
| KCNAB2 | P62482 | 2.0 | 0.92 | X | | | X | |
| NDUFA8 | Q9DCJ5 | 2.0 | 0.96 | X | | | X | |
| NDUFS8 | Q8K3J1 | 2.0 | 1 | | | | X | |
| PDIA6 | Q922R8 | 2.0 | 0.89 | X | | X | | |
| SFXN3* | Q91V61 | 2.0 | 0.44 | X | | | X | |
| SLC25A22 | Q9D6M3 | 2.0 | 0.89 | X | | | X | |
| STMN2 | P55821 | 2.0 | 1 | | | X | | |
| TNR | Q8BYI9 | 2.0 | 1 | X | X | | X | |
| TUBB4B | P68372 | 1.9 | 0.55 | | | | X | X |
| ATP5C1 | Q91VR2 | 1.9 | 0.6 | X | | | X | |
| PPIA | P17742 | 1.8 | 0.74 | | | X | X | |
| CKMT1 | P30275 | 1.8 | 0.86 | X | | X | X | X |
| COX5A | P12787 | 1.8 | 0.54 | | | | X | |
| C1QC* | Q02105 | 1.8 | 0.43 | | | | X | X |
| NDUFS1 | Q91VD9 | 1.8 | 0.88 | X | X | | X | |
| WWP1 | Q8BZZ3 | 1.8 | 0.88 | | | | X | |
| ATP5B | P56480 | 1.7 | 0.38 | X | | X | X | X |
| CCT5 | P80316 | 1.7 | 0.81 | X | | X | X | |
| DCLK1 | Q9JLM8 | 1.7 | 0.96 | X | X | X | | |
| SLC25A3 | Q8VEM8 | 1.7 | 1 | X | X | | X | |
| SPTBN1 | Q62261 | 1.7 | 0.7 | | | X | X | |
| TUBB3 | Q9ERD7 | 1.6 | 0.73 | | | X | X | X |
| CAMK2D | Q6PHZ2 | 1.6 | 0.82 | X | | X | X | |
| ATP6V0A1 | Q9Z1G4 | 1.5 | 0.92 | X | | | X | |
| CFL1* | P18760 | 1.5 | 0.44 | X | | X | X | |
| ATP1A2* | Q6PIE5 | 1.5 | 0.04 | X | | X | X | X |
| ADGRL2 | Q8JZZ7 | 1.5 | 0.89 | | | X | | |
| BSN | O88737 | 1.5 | 0.72 | X | | | X | |
| DBT | P53395 | 1.5 | 0.86 | X | | X | X | |
| GTF2I | Q9ESZ8 | 1.5 | 0.89 | | | | X | |
| HELB | Q6NVF4 | 1.5 | 0.89 | | | | X | |
| HK1 | P17710 | 1.5 | 0.89 | X | | | X | |
| HMCN2 | A2AJ76 | 1.5 | 0.89 | | | | X | |
| LGI3 | Q8K406 | 1.5 | 0.89 | | | | X | |
| PC | Q05920 | 1.5 | 0.89 | | | | X | |
| RAB3IP | Q68EF0 | 1.5 | 0.89 | | | | X | |
| UQCR11 | Q9CPX8 | 1.5 | 0.89 | | | | X | |

Table 1 continued

| Protein name | UniProt ID | Spectral ratio | MaxP | ES | IS | P5 | P50 | pS940 |
|---|---|---|---|---|---|---|---|---|
| ATP1A1 | Q8VDN2 | 1.5 | 0.67 | X | X | | X | X |
| ATP5O | Q9DB20 | 1.5 | 0.71 | | | X | X | |
| NDUFA4 | Q62425 | 1.4 | 0.51 | X | | | X | X |
| ATP6V0D1 | P51863 | 1.4 | 0.53 | X | | | X | |
| ACAT1* | Q8QZT1 | 1.4 | 0.43 | X | | X | X | |
| ATP2B2* | Q9R0K7 | 1.4 | 0.47 | X | | | X | |
| TUBB4A | Q9D6F9 | 1.4 | 0.64 | | | X | X | X |
| GNB1 | P62874 | 1.4 | 0.58 | X | X | X | X | X |
| C1QA | P98086 | 1.4 | 0.36 | | | X | X | |
| NDUFA10 | Q99LC3 | 1.4 | 0.65 | X | | | X | |
| RAP2B* | P61226 | 1.3 | 0.42 | | X | | X | |
| SYT1 | P46096 | 1.3 | 0.65 | X | | | X | X |
| SLC1A3 | P56564 | 1.3 | 0.9 | | X | X | X | |
| CAPRIN1 | Q60865 | 1.3 | 0.56 | | | X | X | |
| YWHAZ* | P63101 | 1.3 | 0.44 | X | X | X | X | X |
| ATP1A3* | Q6PIC6 | 1.3 | 0 | X | | X | X | X |
| STX1B* | P61264 | 1.3 | 0.45 | X | X | | X | X |
| NEFM | P08553 | 1.2 | 0.78 | X | | | X | |
| DLST | Q9D2G2 | 1.2 | 0.72 | X | | X | | |

Orange-Transmembrane.

Grey-Soluble.

Green-Secreted/Extracellular.

ES – excitatory synapse.

IS – inhibitory synapse.

DOI: https://doi.org/10.7554/eLife.28270.017

of N-terminal KCC2b antibodies could shift a proportion of native-KCC2 to higher molecular weights, in comparison to IgY control antibodies (*Figure 5d*). Next, we observed that this antibody-induced shift in native-KCC2b using N-terminal antibody also shifted a population of native-PACSIN1 to comparable higher molecular weights (*Figure 5d*). Collectively, these experiments establish native-PACSIN1 as a novel KCC2-binding partner in whole brain tissue.

The PACSIN family of proteins contains three members that share ~90% amino acid identity (*Modregger et al., 2000*). PACSIN1 is neuron-specific and is broadly expressed across multiple brain regions; PACSIN2 is ubiquitous and is abundant in cerebellar Purkinje neurons (*Anggono et al., 2013*; *Cembrowski et al., 2016*), and PACSIN3 is restricted to muscle and heart (*Modregger et al., 2000*). To determine which members of the PACSIN family binds to KCC2, we transfected PACSIN constructs (*Anggono et al., 2013*), with myc-KCC2b in COS-7 cells and performed co-immunoprecipitation. We observed that KCC2 robustly associates with PACSIN1, weakly interacts with PACSIN2, and does not interact with PACSIN3 (*Figure 5e*). Mouse PASCIN1 contains a membrane-binding F-BAR domain (aa 1–325), a SH3 domain (aa 385–441) that binds to phosphorylated targets, and a VAR (variable) region (aa 326–384) (*Kessels and Qualmann, 2004*; *2015*). To determine the PACSIN1 region that is required for KCC2 binding we repeated our co-immunoprecipitation assays in COS- 7 cells, but this time we used previously characterized PACSIN1 deletion constructs (*Anggono et al., 2013*)(*Figure 5f*). We discovered that removing either the SH3 (lane 3) or the F-BAR region (lane 6) did not disrupt the KCC2:PACSIN1 interaction, indicating that they are not necessary for KCC2 binding. In an analogous result, neither the SH3 domain alone (lane 7) nor F-BAR domain alone (lane 2) could interact with KCC2. However, KCC2 robustly co-precipitated with

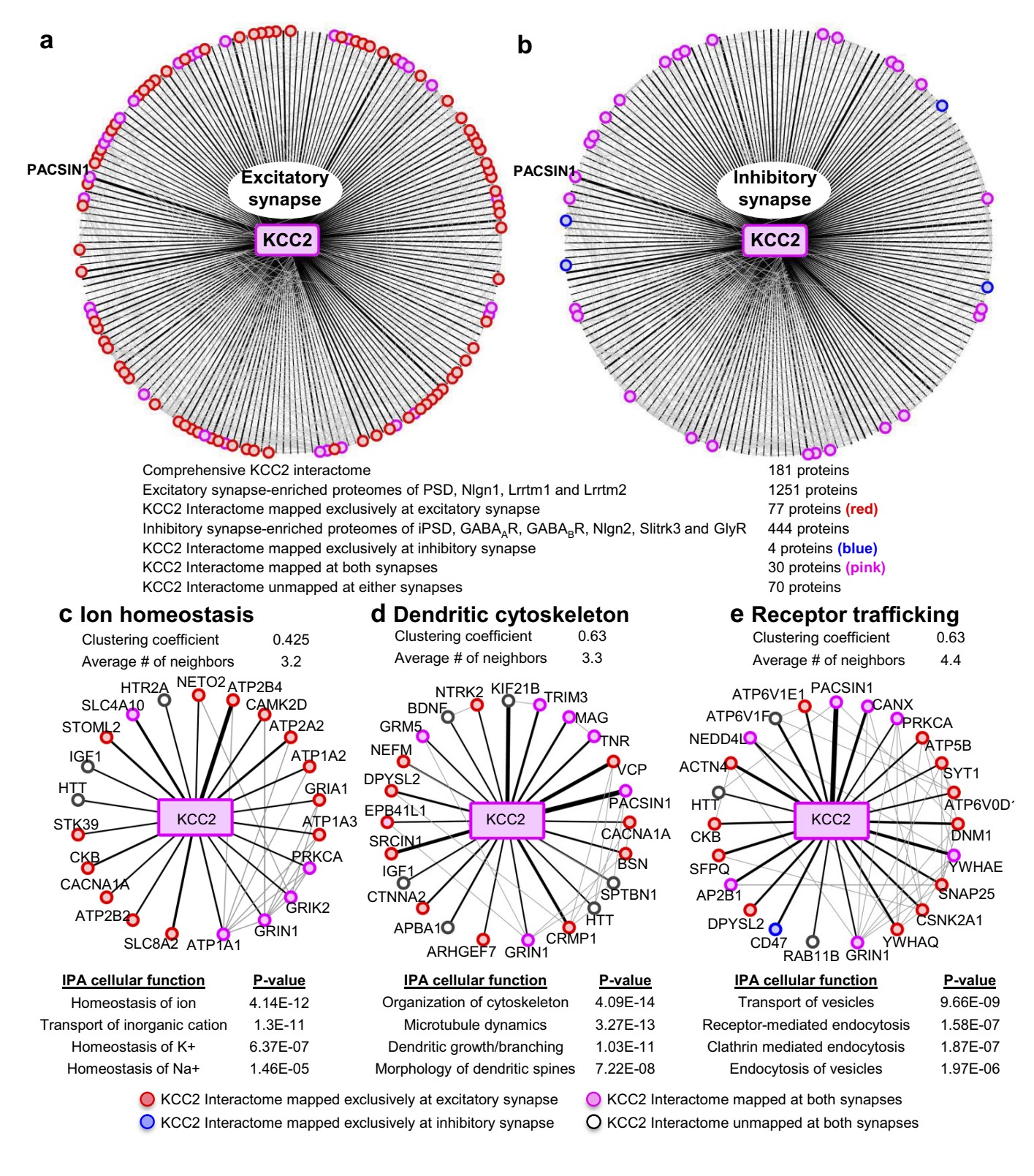

**Figure 4.** Members of the KCC2 interactome are highly represented at excitatory synapses. The KCC2 interactome mapped to (**a**) the excitatory synapse-enriched proteomes or, (**b**) the inhibitory synapse-enriched proteomes. The thickness of the black radial lines in the foreground denotes the number of spectral enrichment (KCC2/IgG) in the log scale. Grey radial lines in the background denotes the previously identified physical/co-expression

*Figure 4 continued on next page*

*Figure 4 continued*

networks across all interactome members. IPA revealing the members of the KCC2 interactome that are involved in (**c**) ion homeostasis, (**d**) dendritic cytoskeleton rearrangement and (**e**) recycling/endocytosis/trafficking. The following source data and figure supplements are available for *Figure 4*.
DOI: https://doi.org/10.7554/eLife.28270.018

The following source data and figure supplement are available for figure 4:

**Source data 1.** Interactome mapping at excitatory and inhibitory synapses.
DOI: https://doi.org/10.7554/eLife.28270.020
**Source data 2.** Ingenuity pathway analysis.
DOI: https://doi.org/10.7554/eLife.28270.021
**Figure supplement 1.** The SLC12A5/KCC2 interactome.
DOI: https://doi.org/10.7554/eLife.28270.019

PACSIN1 when the VAR region alone was co-expressed with KCC2 (lane 5), indicating that the VAR region is sufficient to mediate the KCC2 interaction.

## PACSIN1 is a negative regulator of KCC2 expression and function in hippocampal neurons

KCC2 dysregulation has emerged as a key mechanism underlying several brain disorders including seizures (*Fiumelli et al., 2013*; *Stödberg et al., 2015*; *Saitsu et al., 2016*), neuropathic pain (*Coull et al., 2003*), schizophrenia (*Tao et al., 2012*), and autism spectrum disorders (ASD) (*Cellot and Cherubini, 2014*; *Tang et al., 2016a*). However, there are currently no existing KCC2 enhancers approved for clinical use, and thus there is a critical need to identify novel targets for the development of KCC2 enhancers. To determine whether PACSIN1 may be a potential target for regulating KCC2 function, we assayed for KCC2 function following PACSIN1 knockdown. We chose to assay for the canonical KCC2 function of Cl⁻ extrusion, as the loss of Cl⁻ homeostasis and thus synaptic inhibition, is causal for several neurological disorders (*Coull et al., 2003*; *Huberfeld et al., 2007*; *Tao et al., 2012*; *Cellot and Cherubini, 2014*; *Toda et al., 2014*; *Kahle et al., 2014*; *Puskarjov et al., 2014*; *Stödberg et al., 2015*; *Banerjee et al., 2016*; *Saitsu et al., 2016*; *Tang et al., 2016b*).

We assayed KCC2-mediated Cl⁻ extrusion by performing patch clamp recordings of the reversal potential for GABA ($E_{GABA}$), which is principally determined by $[Cl^-]_i$ (*Kaila, 1994*). Whole-cell patch clamp recordings were obtained from cultured hippocampal neurons that endogenously express KCC2. $E_{GABA}$ was determined from current–voltage (IV) curves that were created by eliciting inhibitory postsynaptic currents (IPSCs), by puffing GABA (20 µM) at the soma while progressively step-depolarizing the postsynaptic holding potential (through current injection via the whole-cell patch pipette). A linear regression of the IPSC amplitude was used to calculate the voltage dependence of IPSCs; the intercept of this line with the abscissa was taken as $E_{GABA}$ (mV), and the slope of this line as GABA conductance (pS). KCC2 function is best determined when the transporter is driven to extrude Cl⁻, which we achieved by loading the intracellular compartment with Cl⁻ via the whole-cell patch pipette (*Doyon et al., 2016*).

To determine whether PACSIN1 regulates KCC2-mediated Cl⁻ transport, we virally transduced neurons with plasmids containing a previously validated PACSIN1 shRNA to knockdown PACSIN1 (*Anggono et al., 2013*), or a scrambled shRNA, which served as the control recordings. Neurons were transduced at 5–7 days in vitro (DIV) and all the recordings were performed at 11–14 DIV. Neurons were selected for recording based on the presence of reporter fluorescence (~60% of neurons were transduced). When neurons expressed the PASCIN1 silencing shRNA, $E_{GABA}$ was significantly hyperpolarized compared to control neurons expressing the scrambled shRNA (*Figure 6a,b*; control shRNA: −28.62 ± 3.07 mV, n = 9; PACSIN1 shRNA: −37.86 ± 1.73 mV, n = 11; t(18)=2.744, p=0.013). We found no significant change in the GABA_AR conductance (*Figure 6a,c*; control shRNA: 6.93 ± 1.32 pS, n = 9; PACSIN1 shRNA: 12.96 ± 2.71 pS, n = 11; t(18)=1.86, p=0.079), which indicates that the effect is on Cl⁻ transport and not GABA conductance. In addition, we found no significant change in the resting membrane potential (RMP), (*Figure 6d*; control shRNA: −63.41 ± 1.48 mV, n = 11; PACSIN1 shRNA: −62.77 ± 2.21 mV, n = 9; t(18)=0.246, p=0.808), which indicates the change in the driving force for Cl⁻ was due to a change in $E_{GABA}$ and not RMP. Together, these data

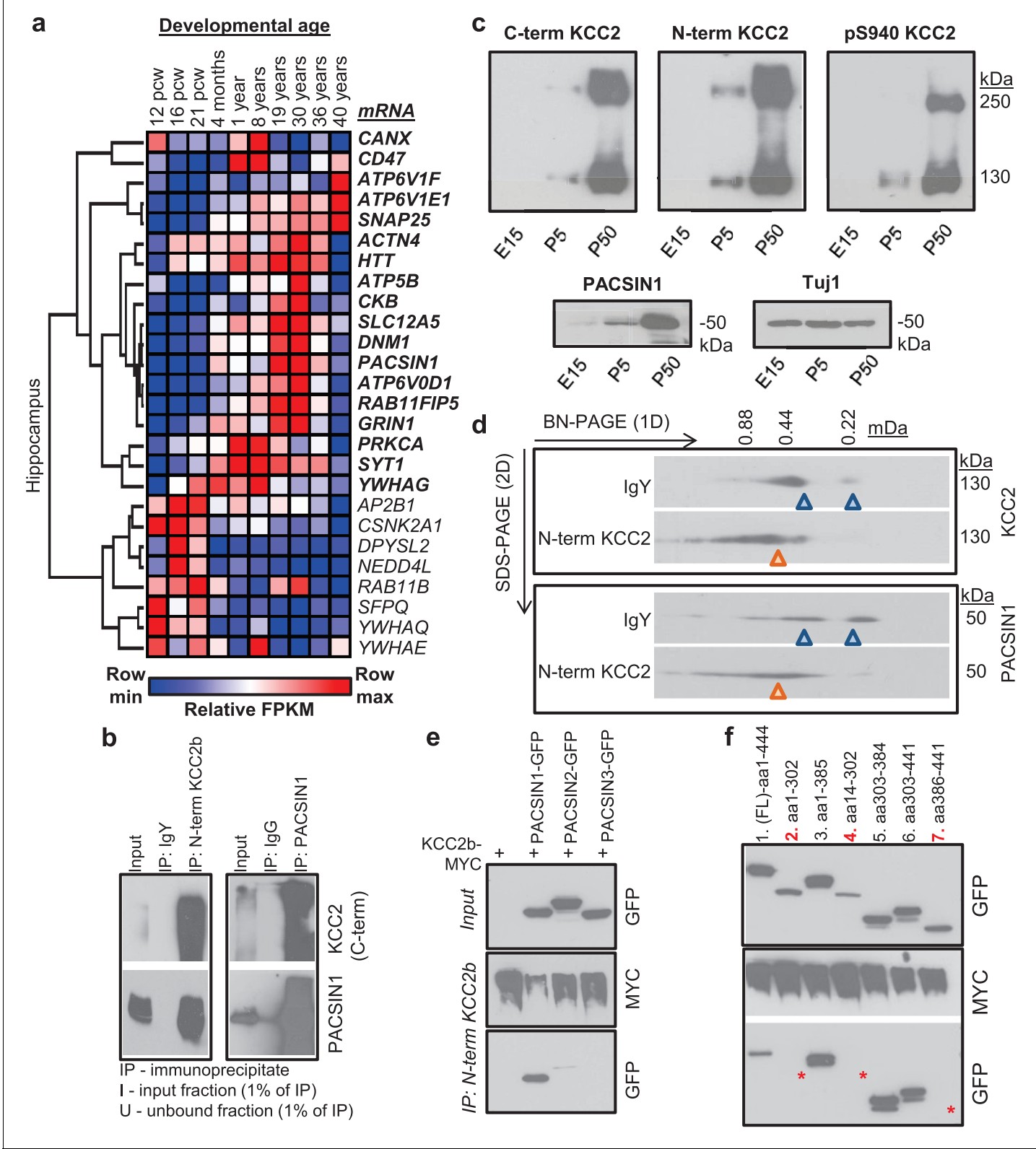

**Figure 5.** Characterization of the PACSIN1-KCC2 interaction. (a) Spatiotemporal expression patterns of *SLC12A5* and members of receptor trafficking node of the KCC2 interactome in the human brain; RNAseq data were analyzed in hippocampus. Pcw, postconceptual weeks. (b) Native KCC2 complexes from $C_{12}E_9$-solubilized whole-brain membrane fractions immunoprecipitated with IgY or anti-N-term KCC2 (left) and IgG or anti-PACSIN1 (right), and immunoblotted with antibodies as indicated. (c) Western blot analysis of developmental expression patterns of KCC2 and PACSIN1. (d)
*Figure 5 continued on next page*

*Figure 5 continued*

Antibody-shift assay followed by 2D-BN-PAGE separation using whole-brain membrane fractions, incubated with antibodies indicated on left; gel separations were immunoblotted with anti-KCC2 or PACSIN1 antibodies. (**e**) Coimmunoprecipitation performed in COS7 cells transfected with myc-tagged KCC2b and GFP-tagged PACSIN1/2/3 constructs, immunoprecipitated with anti-N-term KCC2b, and immunoblotted with the antibodies indicated at right. (**f**) Immunoblot of immunoprecipitates from transfected COS7 cell lysates. * indicate the lanes where PACSIN1 lacks the variable region between ~aa325-383. # of independent biological replicates are indicated in parenthesis: *Figure 5e* (4), *Figure 5f* (3), *Figures 5b,c,d* (2).

DOI: https://doi.org/10.7554/eLife.28270.022

The following source data and figure supplements are available for figure 5:

**Source data 1.** Human brain RNAseq data from the Allan Brain Atlas for receptor trafficking node.

DOI: https://doi.org/10.7554/eLife.28270.025

**Figure supplement 1.** Spatiotemporal expression patterns of *SLC12A5* and members of receptor trafficking node of the KCC2 interactome in the human brain.

DOI: https://doi.org/10.7554/eLife.28270.023

**Figure supplement 2.** The primary amino acid sequence coverage of PACSIN1 (left), and protein coverage of PACSIN1 identified by MS analysis are indicated in red.

DOI: https://doi.org/10.7554/eLife.28270.024

indicate that knocking down PACSIN1 increases KCC2-mediated Cl⁻ transporter, which results in hyperpolarized $E_{GABA}$.

The electrophysiology experiments performed above using Cl⁻ loading through the patch pipette are important to test KCC2 function under an ionic load (*Doyon et al., 2016*). To determine the impact of PACSIN1 on KCC2 function under resting conditions, we repeated the electrophysiology recordings performed above to determine $E_{GABA}$, but this time, we used the gramicidin-perforated patch clamp technique to maintain the neuronal Cl⁻ gradient (*Kyrozis and Reichling, 1995*). Similar to our recordings above, we found a significant hyperpolarizing shift in $E_{GABA}$ in neurons expressing PACSIN1 shRNA versus control cells expressing scrambled shRNA (*Figure 6e*; control shRNA: −46.93 ± 2.78 mV, n = 7; PACSIN1 shRNA: −−72.70 ± 4.70 mV, n = 6; t(11)=4.9115, p=0.0005). Taken together, our whole-cell and gramicidin-perforated patch clamp recordings reveal that knocking down KCC2 in cultured hippocampal neurons leads to a hyperpolarization of $E_{GABA}$ that strengthens inhibition, which results from an increase in KCC2-mediated Cl⁻ extrusion.

Due to the known role of PACSIN1 as an endocytic regulatory protein (*Anggono et al., 2006*; *2013*; *Pérez-Otaño et al., 2006*; *Del Pino et al., 2014*; *Widagdo et al., 2016*), we hypothesized that PACSIN1 negatively regulates KCC2 function by altering its expression in the surface membrane. To test our hypothesis, we performed immunofluorescent staining of endogenous KCC2 in cultured hippocampal neurons transduced with transduced with PACSIN1 shRNA or scrambled shRNA (control cells). We observed a significant increase in KCC2 fluorescence in neurons expressing PASCIN1 shRNA in comparison with controls (*Figure 6e*; control shRNA: 58.86 ± 2.53 A.U., n = 32; PACSIN1 shRNA: 74.05 ± 2.53 A.U., n = 32; t(31)=5.272, p<0.0001). Taken together, our electrophysiological recordings and immunofluorescence results demonstrate that a reduction in PACSIN1 results in increased KCC2 expression and an increase in the strength of inhibition (hyperpolarization of $E_{GABA}$).

If PACSIN1 is a *bona fide* negative regulator of KCC2 expression, then overexpressing PACSIN1 should produce a reduction in KCC2 expression. To test this prediction, we again performed immunofluorescent staining of endogenous KCC2 in cultured hippocampal neurons transduced with either eGFP (control) or PASCIN1-eGFP. We observed a remarkable loss of KCC2 immunofluorescence when PACSIN1 was overexpressed in comparison to control eGFP (*Figure 6f*; control eGFP: 62.1 ± 2.7 A.U., n = 23; PACSIN1-eGFP: 11.31 ± 3.17 A.U., n = 16; t(37)=12.13, p<0.0001), which supports the conclusion that PACSIN1 negatively regulates KCC2 expression.

## Discussion

We determined that the mouse brain KCC2 functional interactome is comprised of 181 proteins; and by mapping the KCC2 interactome to excitatory and inhibitory synapse proteomes and performing ingenuity pathway analysis, we determined that KCC2 partners are highly enriched at excitatory synapses and form a dense network with proteins involved in receptor trafficking. We validated the

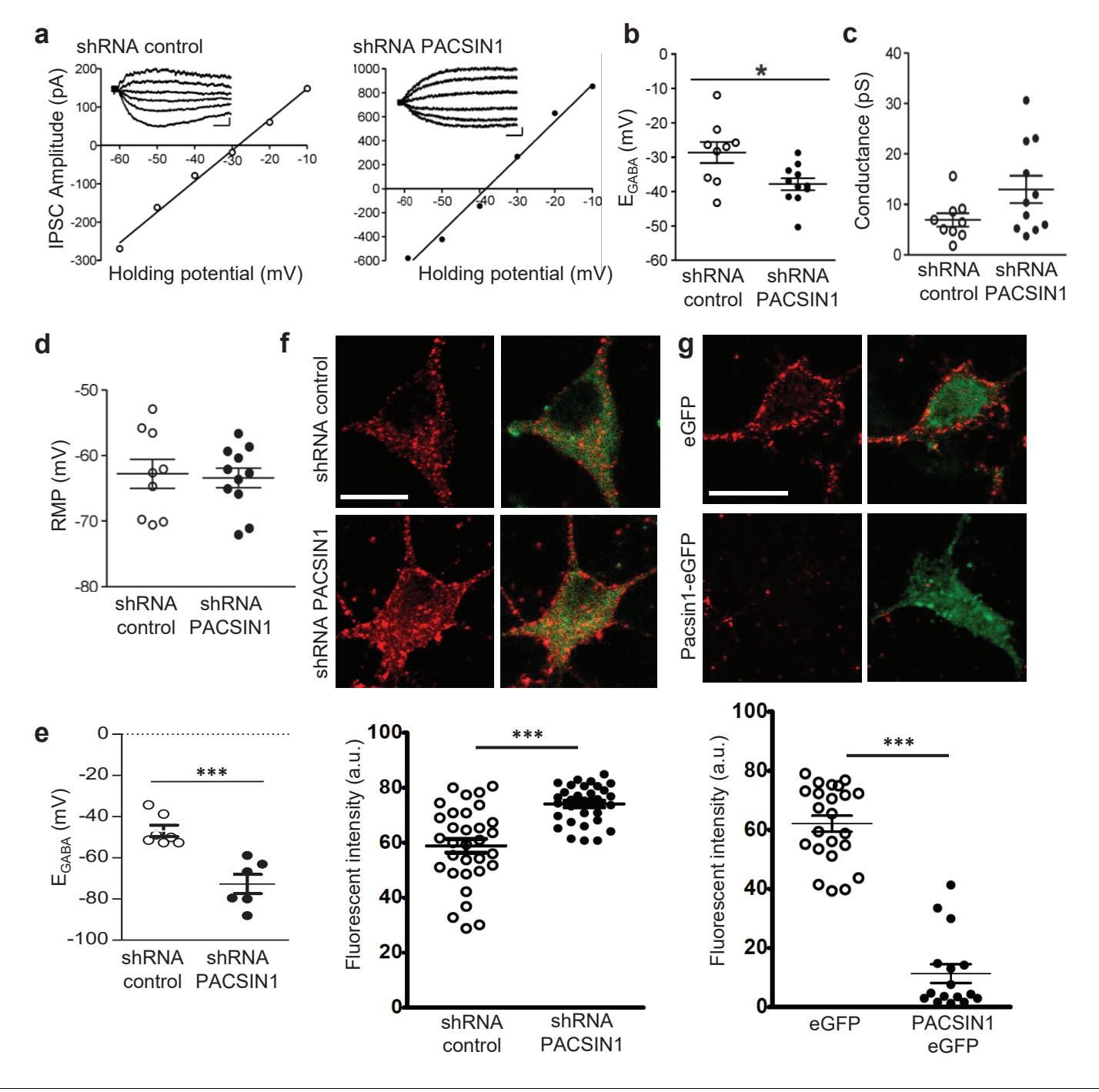

**Figure 6.** PACSIN1 is a negative regulator of KCC2 expression and function. (**a**) Example IV curves measuring $E_{GABA}$ using $Cl^-$-loading through whole-cell configuration from cultured hippocampal neurons. Neurons were transduced with either control shRNA (n = 9; left) or PACSIN1 shRNA (n = 11; right). Summary of (**b**) $E_{GABA}$, (**c**) synaptic conductance, and (**d**) RMP from all experiments similar to the examples in **a**. (**e**) Summary of $E_{GABA}$ recordings obtained by gramicidin-perforated patch clamp recordings. (**f**) Example confocal microscopic immunofluorescent images from cultured hippocampal neurons transduced with control shRNA (n = 32) or PACSIN1 shRNA (n = 32) and stained with anti-KCC2 (red; scale bar, 10 μm); green immunostain reports transfection. Below: summary of fluorescence intensities. (**g**) Similar to f, except neurons were transduced with either control eGFP (n = 23) or PACSIN1-eGFP (n = 16). n values for all experiments on cultured neurons were obtained from a minimum of three independent sets of cultures. Statistical significance was determined using student's t-tests (two-tailed); *p<0.05, ***p<0.001. For all summary plots, the error bars denote mean ± sem. The following figure supplements is available for *Figure 6*.

DOI: https://doi.org/10.7554/eLife.28270.026

*Figure 6 continued on next page*

*Figure 6 continued*

The following figure supplement is available for figure 6:

**Figure supplement 1.** Example illustrating the calculation of fluorescence intensity.
DOI: https://doi.org/10.7554/eLife.28270.027

utility of the KCC2-interactome that we presented, by verifying the biochemical interaction between PACSIN1 and KCC2, and by demonstrating that PACSIN1 participates in regulation of the level of expression of KCC2. Functional validation of the KCC2-PACSIN1 interaction revealed that PACSIN1 robustly and negatively regulates KCC2 expression.

While ion channels and GPCRs are known to predominantly exist in large multi-protein complexes in the CNS (*Husi et al., 2000*; *Berkefeld et al., 2006*; *Collins et al., 2006*; *Müller et al., 2010*; *Schwenk et al., 2010*; *2012*; *2014*; *2016*; *Nakamura et al., 2016*; *Pin and Bettler, 2016*), similar studies on CNS solute carrier proteins (transporters) are still in their infancy (*César-Razquin et al., 2015b*; *Comstra et al., 2017*; *Haase et al., 2017*). Based on the critical importance of SLC transporters as therapeutic targets in both rare and common diseases (*César-Razquin et al., 2015a*; *Lin et al., 2015*), including that of KCC2 in human neurological diseases (*Blaesse et al., 2009*; *Medina et al., 2014*), our present study also fills a general gap in the field of CNS transporter proteomics.

A common caveat of isoform-counting in shot-gun proteomic experiments such as ours, is the problem of protein inference (*Nesvizhskii and Aebersold, 2005*). While we were able to discriminate KCC2 isoforms-a and -b, based on the presence of their unique peptides in their N-terminus (*Figure 1c*), it is not possible to categorize the remaining peptides to either isoform a or b due to their extensive shared homology. Therefore, targeted proteomics such as selected-reaction monitoring would be required to accurately establish the abundances of KCC2 isoforms. We report that native-detergents $C_{12}E_9$ and CHAPS extract KCC2 isoforms differentially (*Figure 1b*). We also report that native-detergents $C_{12}E_9$ and CHAPS pull-down different subsets of proteins along with some common interactors (*Figure 1b*). It is intriguing to note that there were several putative KCC2-partners uniquely identified with CHAPS (SLC44A1, ATP2B4, RNF8, JAGN1, CD47, SLC2A1 (*Figure 1b*). Although we did not perform exhaustive proteomics with CHAPS-based KCC2 extractions, because of the presence of these high-confidence proteins in the CHAPS-based KCC2 LC/MS, we did include these proteins in the KCC2-interactome. This demonstrates that detergent stabilities of KCC2 protein complexes are distinct, in line with other recent ion channel proteomic studies (*Müller et al., 2010*; *Schulte et al., 2011*; *Schwenk et al., 2012*).

Our KCC2 LC/MS identified previously established KCC2 proteins interactors, including ATP1A2 (*Ikeda et al., 2004*), CFL1 (*Chevy et al., 2015*; *Llano et al., 2015*), and CKB (*Inoue et al., 2004*; *2006*), which add confidence to the validity of this interactome. We were initially surprised at the absence of other previously established KCC2 interactors, including: Neto2 (*Ivakine et al., 2013*; *Mahadevan et al., 2015*), GluK2 (*Mahadevan et al., 2014*; *Pressey et al., 2017*), 4.1N (*Li et al., 2007*), beta-pix (*Chevy et al., 2015*; *Llano et al., 2015*), RCC1 (*Garbarini and Delpire, 2008*), or signaling molecules PKC (*Lee et al., 2007*), WNK, SPAK and OSR (*Friedel et al., 2015*). The absence of these previously identified interactors may be due to any of the following caveats, which have been well recognized in previous ion channel and GPCR proteomic studies: (1) these interactions may be weak, transient, mediated by posttranslational modifications (*Schulte et al., 2011*), or mediated by intermediary partners; (2) these interactions are under-represented in the whole brain membrane fractions because they are restricted to specific brain regions; (3) antibody-epitope binding blocked endogenous interactions; (4) despite using the $C_{12}E_9$-based solubilization strategy that is known to stabilize ion pumps and transporters (*Romero, 2009*; *Babu et al., 2010*; *Ramachandran et al., 2013*) particular interactions may be better preserved by other detergent conditions.

Single particle tracking of surface KCC2 has revealed that ~ 66% of KCC2 is located synaptically (*Chamma et al., 2012*, *2013*). While the density of surface KCC2 was not reportedly different between excitatory and inhibitory synapses, KCC2 was shown to dwell longer at excitatory synapses. Our observation that KCC2 interacting proteins are primarily enriched at excitatory synapses in comparison to inhibitory synapses is in line with this increased confinement of KCC2 at excitatory

synapses. The presence of KCC2 at excitatory synapses raises some interesting questions: How does KCC2–mediated Cl⁻ extrusion regulate hyperpolarizing inhibition if it is preferentially localized near excitatory synapses? Why are the KCC2 partners exclusive to the inhibitory synapses, less represented when compared with excitatory synapses? One potential answer to both these questions is that: because of the difficulty in identifying components of inhibitory synapses, our knowledge of the proteins present at these structures is incomplete. Despite the fact that our network mapping incorporated 444 proteins known to be enriched at inhibitory synapses (*Heller et al., 2012*; *Del Pino et al., 2014*; *Kang et al., 2014*; *Loh et al., 2016*; *Nakamura et al., 2016*; *Schwenk et al., 2016*; *Uezu et al., 2016*), it is possible that we identified a smaller representation of inhibitory synapse-specific KCC2 partners in our present study. In this context, it is interesting to note that KCC2 itself was not identified in any of the above inhibitory synapse-enriched proteomes, and we assigned KCC2 as a member of the inhibitory synapse during network analyses, because of its established function at this locus. Another possibility is that KCC2 'moon-lights' between inhibitory and excitatory synapses, as previously suggested (*Blaesse and Schmidt, 2015*). Our interactome supports this hypothesis as we identified 29 proteins (excluding KCC2) that are enriched at both synapses. However, future studies are required to systematically examine whether the KCC2 complexes containing these putative moon-lighting proteins are similar or distinct complexes within these loci. While the notion that excitatory and inhibitory synapses are distinct structures is widely accepted, emerging evidence from cortex suggests this may not be strictly true (*Chiu et al., 2013*; *Higley, 2014*). Under circumstances where excitatory and inhibitory synapses are in close physical proximity, the molecular complex involving KCC2 and these moonlighting proteins are ideally placed to execute cell-intrinsic E/I balance regulation, a hypothesis stemming from our present study that requires rigorous experimental testing.

Ever since the first discovery that KCC2 participates in the regulation of dendritic spine structures (*Li et al., 2007*), several studies have demonstrated 4.1N as a critical mediator of this non-canonical transporter-independent KCC2 function (*Horn et al., 2010*; *Gauvain et al., 2011*; *Chamma et al., 2013*; *Fiumelli et al., 2013*). Recently however, additional molecular players underlying this phenomenon, including COFL1, and ARHGEF7 (Beta-pix) have been identified to interact with KCC2 (*Chevy et al., 2015*; *Llano et al., 2015*). In the present study, we identify diverse high confidence (Platinum, Gold) cytoskeletal organizers belonging to distinct protein families such as CRMPs, 14-3-3 isoforms, SRCIN1, VCP, KIF21B, previously unsuspected to mediate KCC2-dependant non-canonical function. However, the precise relation between KCC2-dependant non-canonical functions and these putative partners in not currently known.

PACSIN1 is a well-established endocytic adapter protein that regulates the surface expression of distinct glutamate (*Anggono et al., 2006*, *2013*; *Pérez-Otaño et al., 2006*; *Widagdo et al., 2016*) and glycine receptors (*Del Pino et al., 2014*). We reveal PACSIN1 as a novel negative regulator of KCC2 expression in hippocampal neurons. We previously reported that native-KCC2 assembles as a hetero-oligomer that migrates predominantly above ~400 kDa (*Mahadevan et al., 2014*; *2015*). Similar to KCC2, native-PACSIN1 also migrates above ~400 kDa (*Kessels and Qualmann, 2006*). Here, we report that while *SLC12A5* and *PACSIN1* mRNA transcripts increase in parallel in multiple brain regions throughout development, PACSIN1 overexpression remarkably decreases total KCC2 abundance. How does PACSIN1 exist in a stable complex with KCC2 when it negatively regulates KCC2 expression? Since KCC2 and PACSIN1 are both dynamically regulated by phosphorylation and PKC (*Anggono et al., 2006*; *Lee et al., 2007*; *Clayton et al., 2009*; *Kahle et al., 2013*), we predict that upon KCC2 phosphorylation, PACSIN1 uncouples from KCC2 rendering it incapable of negatively regulating KCC2. Numerous pathological situations are associated with decreased KCC2 phosphorylation at Ser940 (*Wake et al., 2007*; *Lee et al., 2011*; *Sarkar et al., 2011*; *Toda et al., 2014*; *Ford et al., 2015*; *Mahadevan et al., 2015*; *Silayeva et al., 2015*; *Leonzino et al., 2016*; *Mahadevan and Woodin, 2016*), resulting in decreased transporter expression and/or function. It will be important to determine whether any of these neurological deficits stem from PACSIN1-mediated decreases in KCC2. In the present study, we demonstrate that PACSIN1 shRNA increases KCC2 expression and strengthens inhibition, indicating that PACSIN1 is a target for intervention to upregulate KCC2 during pathological states.

In conclusion, the KCC2 interactome as presented here, serves as a molecular framework for systematically exploring how KCC2 up and down states can be dynamically regulated by its native

molecular constituents, thereby providing a blueprint for subsequent detailed functional investigations.

## Materials and methods

### Animals and approvals

All experiments were performed in accordance with guidelines and approvals from the University of Toronto Animal Care Committee and the Canadian Council on Animal Care (University of Toronto Protocol #20012022). Animals of both sexes from wild-type mice, C57/Bl6 strain (Charles River Laboratories) were used throughout. Animals were housed in the Faculty of Arts and Science Biosciences Facility (BSF) in a 12 hr light: 12 hr d cycle, with 2–5 animals/cage.

### Detergents

All biochemical preparations and centrifugations were performed at 4°C as previously described (*Ivakine et al., 2013*; *Mahadevan et al., 2014*, *2015*). Systematic analysis of detergent solubility, and migration of native-KCC2 from crude membrane fractions were performed according to the workflow described in *Figure 1—figure supplement 1*. The following eight detergents (or detergent combinations) were used to solubilize whole brain membranes: $C_{12}E_9$ (1.5%; nonaethylene glycol monododecyl ether, Sigma-Aldrich, St. Louis, MO, #P9641), CHAPS (1.5%; 3-[(3-Cholamidopropyl) dimethylammonio]−1-propanesulfonate hydrate, Sigma-Aldrich #C3023), DDM, (1.5%; DDM, n-dodecyl β-d-maltoside, Sigma-Aldrich #D4641), DOC (1%; sodium deoxycholate, Sigma-Aldrich #D6750), NP40 (1%; nonyl phenoxypolyethoxylethanol, Thermo Fisher Scientific # 28324), Triton-X-100 (1%; 4-(1,1,3,3-Tetramethylbutyl)phenyl-polyethylene glycol, t-Octylphenoxypolyethoxyethanol, Polyethylene glycol tert-octylphenyl ether, Sigma-Aldrich #X-100), Triton-X-100 (1%) +DOC (1%), SDS (0.1%; Sodium dodecyl sulphate, (Sigma-Aldrich #71725) +DOC (1%)+NP40 0.5%).

### Purification of KCC2 and in vivo co-immunoprecipitation

Mice (~P5, P50) were sacrificed, and brains were removed and homogenized on ice in PBS using a glass-Teflon homogenizer, followed by brief low-speed centrifugation. Soft-pellets were re-suspended in ice-cold lysis buffer [Tris·HCl, 50 mM, pH 7.4; EDTA, 1 mM; protease and phosphatase inhibitor mixture (Roche)], homogenized, and centrifuged for 30 min at 25,000 × g. Membrane pellets were re-suspended in solubilization buffer (4Xw/v) [Tris·HCl, 50 mM, pH 7.4; NaCl, 150 mM; EDTA, 0.05 mM; selected detergent(s), and protease and phosphatase inhibitor mixture(Roche)], solubilized for 3 hr on a rotating platform at 4°C, and centrifuged for 1 hr at 25,000 × g. For KCC2 and control co-immunoprecipitations, 20–100 µl GammaBind IgG beads were incubated on a rotating platform with the following antibodies (5–100 µg antibody) for 4 hr at 4°C in cold 1X PBS:

- mouse polyclonal C-term KCC2 antibody, Neuromab #N1/12, RRID AB_10697875;
- rabbit polyclonal C-term KCC2 antibody, Millipore, Temecula, CA, #07–432, RRID AB_310611;
- rabbit monoclonal pS940 KCC2, Phosphosolutions #p1551-940, RRID AB_2492213;
- chicken polyclonal N-term KCC2b antibody (*Markkanen et al., 2014*);
- IgG/IgY control antibodies

Following antibody binding, 20 mM DMP (dimethyl pimelimidate, ThermoFisher 21667) in cold 1X PBS was used to crosslink antibodies with the beads, according to manufacturer's instructions. The crosslinking reaction was stopped by adding 50 mM Tris·HCl to quench excess DMP, and the antibody-conjugated beads were thoroughly washed with the IP buffer. 1–10 mg of pre-cleared mouse brain membrane fractions were incubated with KCC2 or control antibody-conjugated beads on a rotating platform for 4 hr at 4°C. After co-immunoprecipitation, the appropriate unbound fraction was saved for comparison with an equal amount of protein to calculate the IP-efficiency (*Figure 2—figure supplement 1*). The beads were washed twice with IP buffer containing detergent, and twice with IP-buffer excluding the detergent. The last wash was performed in 50 mM ammonium bicarbonate. Co-immunoprecipitation experiments for validating KCC2 and PACSIN1 was performed similar to the above procedure, in the absence of DMP-crosslinking. In a subset of validation experiments, anti-PACSIN1 antibody (Synaptic Systems #196002, RRID AB_2161839), was used for reverse co-IP. The break-down of LC/MS replicates are as follows:

- Optimization of LC/MS (*Figure 1*) using CHAPS and $C_{12}E_9$-solibilized membrane fractions were performed each with parallel IgG. (6XCHAPS KCC2) + (6XCHAPS IgG) + (2XC12E9 KCC2) + (2XC12E9 IgG)=16 AP/MS using 5 µg C-terminal pan-KCC2 antibody, and 1 mg of P50 membranes.
- LC/MS (*Figures 2* and *3*) using separate $C_{12}E_9$-solibilized membrane fractions (distinct from the lysates used for optimization experiments) were performed (with parallel IgG/IgY) as follows: (1XP50, C-term KCC2) + (1XP50, pS940 KCC2) + (1XP50, IgG) + (1XP50, N-term KCC2b) + (1XP50, IgY) + (1XP5, C-term KCC2) + (1XP5, IgG) + (1XP5, N-term KCC2b) + (1XP5, IgY)=9 AP/MS using 100 µg control/KCC2 antibody, and 10 mg of membrane, constitute the second biological replicate.
- The 25 AP/MS described above constitute the primary data presented in this manuscript. In addition, the 9 AP/MS experiments corresponding to (*Figures 2* and *3*) were repeated again using a separate batch of P5 and P50 lysates distinct from previous replicates, and using 1 mg membrane and 20 ug of Control/KCC2 antibodies, and this constitute the third biological replicate. Presence or absence of a protein across replicates is presented in *Figure 3—source data 3*.

## Mass spectrometry

Mass spectrometry for the creation of the KCC2 interactome (*Figures 2* and *3*) was performed at the SPARC Biocentre at SickKids Research Institute (Toronto, Ontario). Mass spectrometry for the determination of optimal detergents for native KCC2 extraction (*Figure 1*) was performed in the lab of Dr. Tony Pawson at the Lunenfeld-Tanenbaum Research Institute (LTRI), Mount Sinai Hospital (Toronto, ON) and in the CBTC (University of Toronto). Details on the individual experiments performed in each facility is located in *Figure 2—source data 1*.

For all MS experiments, proteins were eluted from beads by treatment with double the bead volume of 0.5 M ammonium hydroxide (pH 11.0), and bead removal by centrifugation; this procedure was repeated 2x. The combined supernatants were dried under vacuum, reduced with DTT, and the free cysteines were alkylated with iodoacetamide. The protein concentration was measured, and trypsin was added at a ratio of 1:50; digestion occurred overnight at 37°C. The peptides were purified by C18 reverse phase chromatography on a ZipTip (Millipore, Bellerica, MA). Specifics of the MS in the three facilities are below:

### SPARC

Orbitrap analyzer (Q-Exactive, ThermoFisher, San Jose, CA) outfitted with a nanospray source and EASY-nLC nano-LC system (ThermoFisher, San Jose, CA). Lyophilized peptide mixtures were dissolved in 0.1% formic acid and loaded onto a 75 µm x 50 cm PepMax RSLC EASY-Spray column filled with 2 µM C18 beads (ThermoFisher San, Jose CA) at a pressure of 800 Bar. Peptides were eluted over 60 min at a rate of 250 nl/min using a 0% to 35% acetonitrile gradient in 0.1% formic acid. Peptides were introduced by nanoelectrospray into the Q-Exactive mass spectrometer (Thermo-Fisher). The instrument method consisted of one MS full scan (400–1500 m/z) in the Orbitrap mass analyzer with an automatic gain control target of 1e6, maximum ion injection time of 120 ms and a resolution of 70,000 followed by 10 data dependent MS/MS scans with a resolution of 17,500, an AGC target of 1e6, maximum ion time of 120 ms, and one microscan. The intensity threshold to trigger a MS/MS scan was set to 1.7e4. Fragmentation occurred in the HCD trap with normalized collision energy set to 27. The dynamic exclusion was applied using a setting of 10 s.

### CBTC

The peptides were analyzed on a linear ion trap-Orbitrap hybrid analyzer (LTQ-Orbitrap, Thermo-Fisher, San Jose, CA) outfitted with a nanospray source and EASY-nLC split-free nano-LC system (ThermoFisher, San Jose, CA). Lyophilized peptide mixtures were dissolved in 0.1% formic acid and loaded onto a 75 µm x 50 cm PepMax RSLC EASY-Spray column filled with 2 µM C18 beads (ThermoFisher San, Jose CA) at a pressure of 800 BAR. Peptides were eluted over 60 min at a rate of 250 nl/min using a 0% to 35% acetonitrile gradient in 0.1% formic acid. Peptides were introduced by nano electrospray into an LTQ-Orbitrap hybrid mass spectrometer (Thermo-Fisher). The instrument method consisted of one MS full scan (400–1500 m/z) in the Orbitrap mass analyzer, an automatic

gain control target of 500,000 with a maximum ion injection of 200 ms, one microscan, and a resolution of 120,000. Ten data-dependent MS/MS scans were performed in the linear ion trap using the ten most intense ions at 35% normalized collision energy. The MS and MS/MS scans were obtained in parallel fashion. In MS/MS mode, automatic gain control targets were 10,000 with a maximum ion injection time of 100 ms. A minimum ion intensity of 1000 was required to trigger an MS/MS spectrum. The dynamic exclusion was applied using a maximum exclusion list of 500 with one repeat count with a repeat duration of 15 s and exclusion duration of 45 s.

### LTRI

Nano-LCMS using a home-packed 0.75 µm x 10cm C18 emitter tip (Reprosil-Pur 120 C18-AQ, 3 µm). A Nano LC-Ultra HPLC system (Eksigent) was coupled to an LTQ Orbitrap Elite (ThermoFisher) and samples were analyzed in data-dependent acquisition mode. A 60000 resolution MS scan was followed by 10 CID MS/MS ion trap scans on multiply charged precursor ions with a dynamic exclusion of 20 s. The LC gradient was delivered at 200 nl/minute and consisted of a ramp of 2–35% acetonitrile (0.1% formic acid) over 90 min, 35–80% acetonitrile (0.1% formic acid) over 5 min, 80% acetonitrile (0.1% formic acid) for 5 min, and then 2% acetonitrile for 20 min.

## Analysis of mass spectra and protein identification

All MS/MS samples were analyzed using Sequest (Thermo Fisher Scientific, San Jose, CA, USA; version 1.4.0.288) and X! Tandem (The GPM, thegpm.org; version CYCLONE (2010.12.01.1)). Sequest was set up to search Uniprot-mus +musculus_reviewed_Oct172015.fasta (unknown version, 25231 entries) assuming the digestion enzyme trypsin. X! Tandem was set up to search the Uniprot-mus +musculus_reviewed_Oct172015 database (25248 entries) also assuming trypsin. Sequest and X! Tandem were searched with a fragment ion mass tolerance of 0.020 Da and a parent ion tolerance of 10.0 PPM. Carbamidomethyl of cysteine was specified in Sequest and X! Tandem as a fixed modification. Deamidated of asparagine and glutamine and oxidation of methionine were specified in Sequest as variable modifications. Glu->pyro Glu of the n-terminus, ammonia-loss of the n-terminus, gln->pyro Glu of the n-terminus, deamidated of asparagine and glutamine and oxidation of methionine were specified in X! Tandem as variable modifications.

Scaffold (version Scaffold_4.7.2, Proteome Software Inc., Portland, OR) was used to validate MS/MS-based peptide and protein identifications. Peptide identifications were accepted if they could be established at greater than 95.0% probability. Peptide Probabilities from X! Tandem were assigned by the Peptide Prophet algorithm (*Keller et al., 2002*) with Scaffold delta-mass correction. Peptide probabilities from Sequest were assigned by the Scaffold Local FDR algorithm. Protein identifications were accepted if they could be established at greater than 95.0% probability and contained at least one identified peptide. Protein probabilities were assigned by the Protein Prophet algorithm (*Nesvizhskii et al., 2003*). Proteins that contained similar peptides and could not be differentiated based on MS/MS analysis alone were grouped to satisfy the principles of parsimony. Proteins were annotated with GO terms from gene_association.goa_uniprot (downloaded Dec 14, 2015) (*Ashburner et al., 2000*).

In addition, peak lists obtained from MS/MS spectra were identified independently using OMSSA version 2.1.9 (*Geer et al., 2004*), X!Tandem version X! Tandem Sledgehammer (2013.09.01.1) (*Craig and Beavis, 2004*), Andromeda version 1.5.3.4 (*Cox et al., 2011*), MS Amanda version 1.0.0.5242 (*Dorfer et al., 2014*), MS-GF +version Beta (v10282) (*Kim and Pevzner, 2014*), Comet version 2015.02 rev. 3 (*Eng et al., 2013*), MyriMatch version 2.2.140 (*Tabb et al., 2007*) and Tide (*Diament and Noble, 2011*). The search was conducted using SearchGUI version 2.2.2 (*Vaudel et al., 2011*).

Protein identification was conducted against a concatenated target/decoy version (*Elias and Gygi, 2010*) of the Mus musculus (24797,>99.9%), Sus scrofa (1,<0.1%) complement of the UniProtKB (*Apweiler et al., 2004*) (version of December 2015, 24798, Mus Musculus) canonical and isoform sequences). The decoy sequences were created by reversing the target sequences in SearchGUI. The identification settings were as follows: Trypsin with a maximum of two missed cleavages; 10.0 ppm as MS1 and 0.5 Da as MS2 tolerances; fixed modifications: Carbamidomethylation of C (+57.021464 Da), variable modifications: Deamidation of N (+0.984016 Da), Deamidation of Q (+0.984016 Da), Oxidation of M (+15.994915 Da), Pyrolidone from E (-—18.010565 Da), Pyrolidone

from Q ($-$$-$17.026549 Da), Pyrolidone from carbamidomethylated C ($-$$-$17.026549 Da) and Acetylation of protein N-term (+42.010565 Da), fixed modifications during refinement procedure: Carbamidomethylation of C (+57.021464 Da).

Peptides and proteins were inferred from the spectrum identification results using PeptideShaker version 1.9.0 (*Vaudel et al., 2015*). Peptide Spectrum Matches (PSMs), peptides and proteins were validated at a 1.0% False Discovery Rate (FDR) estimated using the decoy-hit distribution. Spectrum counting abundance indexes were estimated using the Normalized Spectrum Abundance Factor (*Powell et al., 2004*) adapted for better handling of protein inference issues and peptide detectability. While the two independent protein algorithm searches largely matched with each other, a small subset of proteins were identified with high confidence using the SearchGUI/Peptideshaker platforms that were not identified with the ThermoFisher Scientific/Scaffold platforms.

The mass spectrometry data along with the identification results have been deposited to the ProteomeXchange Consortium (*Vizcaíno et al., 2014*) via the PRIDE partner repository (*Martens et al., 2005*) at https://www.ebi.ac.uk/pride/archive/ with the dataset identifier PXD006046.

## Dataset filtering

Protein candidates from replicate LC/MS screens were subject to the following criteria to build the KCC2 interactome. First pass filter grouping: at least two unique peptides and fold change of total spectra above 1.5. Second pass filter grouping: for proteins with only one unique peptide, consider whether (a) the protein isoform is an already validated KCC2 interactor in literature; (b) the protein isoform already appears in the first pass filter; (c) the protein isoform appears as a single-peptide interactor across experiments (using the same epitope KCC2 IPs/different epitope KCC2 IPs/different developmental time KCC2 IPs). If a particular protein isoform matches any of the above criteria, it gets shifted to the first pass filter grouping. Finally, the proteins that appear in the KCC2 interactome that are previously identified spurious interactors as identified in the CRAPome database (*Mellacheruvu et al., 2013*) were further eliminated. For the existing proteins, a MaxP-SAINT score (*Choi et al., 2012*) was assigned and proteins were grouped as Platinum, Gold, Silver or Bronze interactors prior to subsequent PPI (protein-protein interaction) network analysis. See Figure 3—figure supplement 3 for a detailed description of the path towards constructing the KCC2 interactome.

## Integrated PPI network analysis and data representation

Protein interactions were integrated with curated, high-throughput and predicted interactions from I2D ver. 2.3 database (*Brown and Jurisica, 2007*), FpClass high-confidence predictions (*Kotlyar et al., 2015*) and from the BioGRID database (*Stark et al., 2006*). Networks were visualized using Cytoscape ver. 3.3.0 (*Shannon et al., 2003*; *Cline et al., 2007*). Components of the KCC2 interactome were mapped to the excitatory synapse-enriched PSD, Nlgn1, Lrrtm1 and Lrrtm2 proteomes (*Collins et al., 2006*; *Loh et al., 2016*), or the inhibitory synapse-enriched GABA$_A$R, GABA$_B$R, Nlgn2, Slitrk3 and GlyR proteomes (*Heller et al., 2012*; *Del Pino et al., 2014*; *Kang et al., 2014*; *Loh et al., 2016*; *Nakamura et al., 2016*; *Schwenk et al., 2016*; *Uezu et al., 2016*). In the PPI networks, the thickness of the black radial lines in the foreground denotes the number of spectral enrichment (KCC2/IgG) in the log scale ranging from 2.13 for the highly enriched interactor to 0.08 for the least enriched interactor. For representing the previously established KCC2 physical/functional interactors not identified in this study, an arbitrary value of 0.05 was used for indicating the thickness of black radial lines (See *Figure 4—source data 1*). Grey radial lines in the PPI network background denotes the previously identified physical/co-expression networks across all interactome members as identified from BioGRID and FpClass databases. Venn diagrams were made using Venny, online tool (http://bioinfogp.cnb.csic.es/tools/venny/); heat maps were made using Morpheus online tool provided by the Broad Institute (https://software.broadinstitute.org/morpheus/).

## In vitro co-immunoprecipitation

HEK-293 and COS7 cells obtained from the ATCC were authenticated using Short Tandem Repeat (STR) profiling and checked for mycoplasma contamination. For co-immunoprecipitation experiments, cells were transfected with KCC2b-MYC, eGFP control, eGFP-PACSIN1/2/3, or eGFP-PACSIN1-deletion constructs (0.25 μg/construct) using Lipofectamine (Invitrogen) at 70% confluency. Thirty-six hours after transfection, cells were washed with ice-cold 1 $\times$ PBS and lysed in modified

RIPA buffer [50 mM Tris·HCl, pH 7.4, 150 mM NaCl, 1 mM EDTA, 1% Nonidet P-40, 0.1% SDS, 0.5% DOC, and protease inhibitors (Roche)]. Lysed cells were incubated on ice for 30 min and were centrifuged at 15,000 × g for 15 min at 4°C. Cell lysates or solubilized membrane fractions (~0.2–0.5 mg protein) were incubated with N-terminal KCC2b or anti-myc (CST #9B11, RRID AB_331783) antibodies on a rotating platform (4 hr, 4°C). Lysates were subsequently incubated with 20 µl GammaBind IgG beads (GE Healthcare) on a rotating platform (1 hr at 4°C). After incubation, beads were washed twice with modified RIPA buffer, and twice with modified RIPA buffer minus detergents. Bound proteins were eluted with SDS sample buffer and subjected to SDS/PAGE along with 10% of input fraction and immunoblotted. *Figure 5e* is representative of four independent biological replicates; *Figure 5f* is representative of three independent biological replicates.

## BN-PAGE analysis and antibody-shift assay

Native-membrane fractions were prepared similarly as described (*Swamy et al., 2006*; *Schwenk et al., 2012*; *Mahadevan et al., 2014*; *2015*). Antibody-shift assay and 2D BN-PAGE analysis of native-KCC2 complexes were performed as described previously (*Mahadevan et al., 2014*; *2015*). Briefly, 50 µg - 100 µg of $C_{12}E_9$ solubilized complexes were pre-incubated for 1 hr with 10 µg of anti-N-terminal KCC2b antibody or chicken IgY whole molecule, prior to the addition of Coomassie blue G250. 1D-BN-PAGE was performed as described above using home-made 4% and 5% bistris gels as described (*Swamy et al., 2006*). After the completion of the gel run, excised BN- PAGE lanes were equilibrated in Laemmli buffer containing SDS and DTT for 15 min at room temperature to denature the native proteins. After a brief rinse in SDS- PAGE running buffer, the excised BN-PAGE lanes were placed on a 6% or 8% SDS- PAGE gel for separation in the second dimension. After standard electro-blotting of SDS- PAGE-resolved samples on nitrocellulose membrane, the blot was cut into two molecular weight ranges; the top blots were subjected to western blotting analysis with Rb anti- KCC2, and the bottom blots with Rb anti-PACSIN1. Antibody-shift experiments (*Figure 5d*) using hippocampal membranes are representative from two independent biological replicates.

## PACSIN overexpression and shRNA constructs

All PACSIN constructs used for overexpression and shRNA constructs have been previously validated for specificity (*Anggono et al., 2013*; *Widagdo et al., 2016*). The PACSIN1 shRNA-targeting sequence (sh#1, 5′-GCGCCAGCTCATCGAGAAA-3′) or control shRNA sequence was inserted into the pSuper vector system (Oligoengine) as described previously (*Anggono et al., 2013*). The efficiency and specificity of the PACSIN1 and control shRNA constructs were tested in HEK 293 T cells overexpressing GFP-PACSIN1, and they were subsequently cloned into pAAV-U6 for lentiviral production (serotype AAV2/9).

## Cultured hippocampal neurons and electrophysiology

Low-density cultures of dissociated mouse hippocampal neurons were prepared as previously described (*Acton et al., 2012*; *Mahadevan et al., 2014*). Electrophysiological recordings were performed using pipettes made from glass capillaries (World Precision Instruments, Sarasota, FL), as previously described (*Acton et al., 2012*; *Mahadevan et al., 2014*). Neuronal transduction with viral vectors was performed at DIV 5–7 and the recordings were performed at DIV 11–14. For $Cl^-$ loading experiments in whole-cell configuration, pipettes (5–7 MΩ) were filled with an internal solution containing the following (in mM): 110 $K^+$-gluconate, 30 KCl, 10 HEPES, 0.2 EGTA, 4 ATP, 0.3 GTP, and 13 phosphocreatine (pH 7.4 with KOH, 300 mOsm). For gramicidin-perforated recordings, pipettes were filled with an internal solution containing the following (in mM): 130 $K^+$-gluconate, 10 KCl, 10 HEPES, 0.2 EGTA, 4 ATP, 0.3 GTP, 13 phosphocreatine, and 50 µg/ml gramicidin (pH 7.4 with KOH, 300 mOsm). Cultured neurons were continuously perfused (at ~1 mL/min) with standard extracellular solution containing: 150 NaCl, 3 KCl, 3 $CaCl_2 \cdot 2H_2O$, 2 $MgCl_2 \cdot 6H_2O$, 10 HEPES, and 5 glucose (pH 7.4 with NaOH, 300 mOsm). Cultured neurons were selected for electrophysiology based on the following criteria: (1) a healthy oval or pyramidal-shaped cell body; (2) multiple clearly identifiable processes; (3) a cell body and proximal dendrites that were relatively isolated; and (4) reporter fluorescence (if applicable). Recordings started when the series resistance dropped below 50 MΩ. Recordings were amplified with an Axon Instruments Multiclamp 700B and digitized using an Axon

Instruments Digidata 1322a (Molecular Devices; Sunnyvale, CA). To determine the reversal potential for GABA ($E_{GABA}$), neurons were held at −60 mV under whole-cell voltage clamp and the membrane potential was stepped in +10 mV increments from −80 to −40 mV (this holding potential was set through current injection via the whole-cell patch pipette). During each membrane potential step, a 20 µM GABA puff was applied onto the soma using a picospritzer (Parker, Hollis, NH, USA). The IPSC amplitude represents the maximum current measured during the recordings performed for the $E_{GABA}$ measurement. Using Clampfit (version 9.2; Molecular Devices; Sunnyvale, CA, USA), two cursors were placed on the recording trace (one just before the current, and the other at the peak of the current), and the peak amplitude was exported to Prism (version 5.01) and graphed against the holding potential. A linear regression of the IPSP amplitude versus membrane/holding potential was achieved using Prism and the intercept of this line with the abscissa was taken as $E_{GABA}$, and the slope of this line was taken as the synaptic conductance. For resting membrane potential, a whole-cell patch clamp was achieved, the amplifier was set to I = 0, and the corresponding potential was measured under current clamp mode. Electrophysiological values have not been corrected for the liquid junction potential of ~7 mV.

## Fixed immunostaining and confocal microscopy

DIV 12–14 cultured hippocampal neurons with were first rinsed with 1X PBS, and fixed in 4% paraformaldehyde for 10 min on ice followed by washing thrice with 1X PBS. Neurons were then permeabilized with 1X PBS containing 10% goat serum and 0.5% Triton X-100 for 30 min, followed by a 45 min incubation with rabbit anti-KCC2 (Millipore 07–432) antibodies at 37°C to detect endogenous proteins. Finally, neurons were washed thrice with 1X PBS and incubated with Alexa-fluor 555-conjugated goat anti-rabbit antibody for 45 min at 37°C. Neurons were imaged on a Leica TCS SP8 confocal system with a Leica DMI 6000 inverted microscope (Quorum Technologies). Cultured neurons were selected for immunostaining based on the following criteria: (1) with a healthy oval or pyramidal-shaped cell body; (2) multiple clearly identifiable processes; (3) a cell body and proximal dendrites that were relatively isolated; (4) reporter fluorescence (if applicable). Images were acquired using 3D Image Analysis software (Perkin Elmer). Images were obtained using a 63 × 1.4 NA oil immersion objective. Imaging experiments were performed and analyzed in a blinded manner. Using ImageJ, four bisecting lines were drawn across the center of the cell (*Figure 6—figure supplement 1*). The peak values of each line (2 values/line) were used to calculate peak fluorescent intensity of KCC2 at the membrane. Fluorescence intensity is plotted in arbitrary units (a.u.).

## Statistics

For electrophysiology and immunostaining data (*Figure 6*), 'n' values report the number of neurons, and were obtained from a minimum of three independent sets of cultured neurons (produced from different litters). Example recordings in *Figure 6a* are representative of n = 9 (shRNA control) and n = 11 (PACSIN1 shRNA). Example recordings in *Figure 6e* are representative of n = 32 (shRNA control) and n = 32 (PACSIN1 shRNA). Example recordings in *Figure 6f* are representative of n = 23 (eGFP) and n = 16 (PACSIN1-eGFP). Data in *Figure 6b,c,e and f* are mean ± SEM. Statistical significance was determined using either SigmaStat or GraphPad Prism (version 5.01) software. Statistical significance in *Figure 6b,c,d,e and f* was determined using Student's t-tests (two-tailed); all data sets passed the normal distribution assumptions test. Statistical significance is noted as follows: *p<0.05, **p<0.01, ***p<0.001. Exact p and t values are reported in the Results text.

*Figures 1a*, *5b, c and d* are representative of two independent biological replicates. *Figure 5f* is representative of three independent biological replicates. *Figure 5e* is representative of four independent biological replicates.

## Acknowledgements

We thank the following persons for technical assistance with LC/MS: Dr. Suzanne Ackloo at the Centre for Biological Timing and Cognition, University of Toronto; Drs. Paul Taylor and Jonathan Krieger, at the SPARC BioCentre, Hospital for Sick Children. We thank Dr. Karen Colwill at the Lunenfeld-Tanenbaum Research Institute, Mount Sinai Hospital for coordinating data sharing and LC-MS analyses from experiments conducted at the LTRI facility. We thank EMBL-EBI/Wellcome Trust Proteomics Bioinformatics workshop, Drs. Harald Barsnes and Marc Vaudel, University of

Bergen, for assistance with the PeptideShaker/CompOmics applications. This study was funded by the following: Simons Foundation Autism Research Initiative to MW and YDK,; CIHR training grant from the Sleep and Biological Rhythms Training Program, Toronto to VM; The Academy of Finland grants to P.U. and MSA; The John T. Reid Charitable Trusts and Australian Research Council (ARC) project grant (DP170102402) to VA; CIHR foundation grant FRN-133431 to AE; CIHR Operating Grant to MW.

## Additional information

### Funding

| Funder | Author |
| --- | --- |
| Canadian Institutes of Health Research | Vivek Mahadevan<br>Andrew Emili<br>Melanie A Woodin |
| Simons Foundation | Yves De Koninck<br>Melanie A Woodin |

The funders had no role in study design, data collection and interpretation, or the decision to submit the work for publication.

### Author contributions

Vivek Mahadevan, Conceptualization, Data curation, Formal analysis, Investigation, Methodology, Writing—original draft, Writing—review and editing; C Sahara Khademullah, Zahra Dargaei, Jonah Chevrier, Julian Kwan, Richard D Bagshaw, Investigation, Methodology; Pavel Uvarov, Resources, Validation, Investigation, Methodology; Tony Pawson, Matti Airaksinen, Resources, Supervision, Methodology; Andrew Emili, Resources, Software, Supervision, Methodology; Yves De Koninck, Supervision, Funding acquisition; Victor Anggono, Resources, Methodology; Melanie A Woodin, Conceptualization, Supervision, Funding acquisition, Writing—original draft, Project administration, Writing—review and editing

### Author ORCIDs

Vivek Mahadevan, http://orcid.org/0000-0002-0805-827X
Pavel Uvarov, http://orcid.org/0000-0001-8439-6150
Victor Anggono, http://orcid.org/0000-0002-4062-4884
Melanie A Woodin, https://orcid.org/0000-0003-2984-8630

### Ethics

Animal experimentation: All experiments were performed in accordance with guidelines and approvals from the University of Toronto Animal Care Committee and the Canadian Council on Animal Care (University of Toronto Protocol #20012022). Animals of both sexes from wild-type mice, C57/Bl6 strain (Charles River Laboratories) were used throughout. Animals were housed in the Faculty of Arts and Science Biosciences Facility (BSF) in a 12h light: 12h d cycle, with 2-5 animals/cage.

### Decision letter and Author response

Decision letter https://doi.org/10.7554/eLife.28270.031
Author response https://doi.org/10.7554/eLife.28270.032

## Additional files

### Supplementary files

• Transparent reporting form
DOI: https://doi.org/10.7554/eLife.28270.028

### Major datasets

The following dataset was generated:

| Author(s) | Year | Dataset title | Dataset URL | Database, license, and accessibility information |
|---|---|---|---|---|
| Mahadevan V, Woodin MA | 2017 | KCC2 Interactome | https://www.ebi.ac.uk/pride/archive/projects/PXD006046 | Publicly available at EBI PRIDE Archive (accession no: PXD006046) |

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
