## [Decision Letter]

Thank you for submitting your article "Native KCC2 interactome reveals PACSIN1 as a critical regulator of synaptic inhibition" for consideration by *eLife*. Your article has been reviewed by three peer reviewers and the evaluation has been overseen by a Reviewing Editor and Eve Marder as the Senior Editor. The following individuals involved in review of your submission have agreed to reveal their identity: Mary B Kennedy (Reviewer #1) and Kristopher Kahle (Reviewer #3).

The reviewers have discussed the reviews with one another and the Reviewing Editor has drafted this decision to help you prepare a revised submission.

Summary:

The study by Mahadevan et al., presents the assembly of an unbiased "interactome", or set of proteins that interact with the potassium/chloride ion transporter (KCC2) in neurons. The interactome was determined by identifying proteins co–precipitated with KCC2 by affinity purification with three different antibodies recognizing three different epitopes. The identification was carried out by high resolution mass spectrometry. Co–precipitated proteins were ranked roughly according to their abundance (spectral counts) and consistent appearance in the antibody–bound precipitate. The authors then used proteomic data analytical methods to show the distribution of associated proteins between excitatory and inhibitory synapses; making the somewhat unexpected finding that many more of the KCC2 associated proteins are associated with excitatory synapses than inhibitory synapses, based on various annotated databases. They used the GO database to show that proteins that interact with KCC2 fall into three functional groups; ion homeostasis, dendritic cytoskeleton, and receptor trafficking. An unexpected finding is that the largest and most densely connected such group is receptor trafficking, and the most consistently abundant interactor, measured by spectral counts, is the protein termed pacsin, which has been linked to regulation of receptor endocytosis.

The authors show convincingly with biochemical experiments that the interaction between KCC2 and pacsin is direct and involves the VAR region of pacsin. Finally, they employ electrophysiological methods to show that downregulation of pacsin by shRNA results in upregulation of the amount of KCC2 and overexpression of pacsin results in a dramatic decrease in the expression of KCC2 in neurons. As expected, down regulation of pacsin by shRNA leads to an increase in chloride exclusion from neurons and a shift in the reversal potential for GABA transmission. This, along with their catalogue of the global native–KCC2 interactome, will be a tremendous resource for investigators in this field.

Essential revisions:

The manuscript is, for the most part well–written and convincing. The study is a clear example of an unbiased "global" proteomic study leading to discovery of a new and potentially clinically relevant regulatory mechanism. However, the presentation of the electrophysiological and imaging experiments was far too terse and included some errors. This led to quite a bit of confusion among the reviewers. The authors must extensively edit and expand the electrophysiology sections of the manuscript and better explain the rationale for their experiments as well as the details of how the experiments were carried out.

Following are revisions that must be made to increase the accessibility of the manuscript:

1) In the Materials and methods section, the authors must include more detail about the design and timing of the electrophysiology experiments. At what age were the neurons transfected? In each case, how long after the transfection were the recordings made? List explicitly the composition of the extracellular solution. For Figure 6 were the holding potentials set through the whole cell pipette? At what point in the IPSC was the amplitude measured? In the Results section discussing Figure 6 paragraph discussing each experiment should begin with the purpose of the experiment, carefully explain the design of the experiment, then end with a statement of what the results of the experiment mean. For panel 6C, fix the discrepancy between the units in the figure (pS), and the statement in subsection “PACSIN1 is a negative regulator of KCC2 expression and function in hippocampal neurons”, which lists the results as mV. Provide a separate paragraph explaining the rationale, execution, and conclusions from the gramicidin experiment. What were the "control cells" for this experiment? What were the chloride concentrations applied for control and gramicidin cells? Why is EGABA for control cells –10 mV; whereas in the shRNA control cells, the EGABA was –30 mV? An experiment must be performed in which the gramicidin perforated–patch recordings for the identification of the chloride reversal potential in the PACSIN1 shRNA KD neurons are compared with gramicidin perforated patch recordings in neurons expressing non–sense shRNA.

2) For Figure 6, provide information in the Materials and methods section about how the "fluorescence intensity was calculated. Do these numbers reflect total intensity over the whole field at each wavelength? Or were the images segmented in some way?

3) Although no significant change was observed in Figure 6, that there was a trend to larger GABA chloride conductance's in the pacsin KD neurons raises the question whether the KD of PACSIN1 may potentially have influenced the expression of other chloride channels. More importantly, the chloride reversal potential depends not only on the chloride conductance but also on the membrane potential. The authors should provide information about the resting potentials in the control and KD neurons.

With respect to the proteomics experiments, several clarifications are needed.

1) In the Materials and methods section, it is stated that the experiments providing data shown in Figure 2 and Figure 3 were performed only once. These experiments should be repeated at least one more time.

2) In subsection “Determining Affinity Purification (AP) conditions to extract native–KCC2”, the chemical names of the three detergents (C12E9, CHAPS, and DDM) should be given at this first use of their abbreviated names. Similarly, in the Materials and methods section, the sources of each of the listed detergents should be given (i.e. where they were purchased).

3) Early in the manuscript, the relationship between the terms KCC2a, KCC2b, and SLC12A5 should be clearly stated. Is SLC12A5 the name of the gene that encodes KCC2a and b? If so, that term shouldn't be listed under the heading of Protein ID in Table 1. Should the heading instead by Gene name? It is a confusing practice to conflate the name given to a gene with the name of the protein. It leads to poor communication across research fields.

4) Similarly, in subsection “Members of KCC2 interactome are highly represented at excitatory synapses”, the meaning of SLC12A5 mRNA should be clarified. Does this term include mRNA encoding both the KCC2 isoforms? (Note that the word "members" is duplicated.)

5) The legend for Figure 4 is incorrect and appears incomplete. The circles in 4A are red not pink. The meaning of the blue circles in 4B should be stated. The legend states that the thickness of the edge denotes the number of spectral enrichment, however, it would be clearer to say the thickness of the "radial line" or some other term since the lines in question don't appear as an edge. Upon close examination, one can see that the radial lines and the very light grey lines in 4A and 4B are identical. However, in 4A the grey lines appear to pass over the radial lines; whereas, in 4B they are behind the radial lines. It would be better to make the radial and background lines in the two figures identical so that the only differences are the red and blue circles denoting association with excitatory or inhibitory synapses. The explanation of the depiction of the mapping in these two figures should be elaborated in the figure legend. It is too terse, and makes the figure difficult to understand. Similarly, the listing of numbers of proteins in various groups that appears underneath the two circles in 4A and 4B is unexplained and seems out of place. It would be better to list those numbers in the figure legend, in association with a more complete description of the circular depictions.

6) The legend of Figure 5, or the description of Figure 5, subsection “PACSIN1 is a novel native–KCC2 binding partner” should give the amino acid residue boundaries of the F–bar, SH3, and VAR regions in pacsin.

7) The left side of Figure 5 appears to be mislabeled with IgG that should be IgY according to the legend.

8) In the Discussion section, the authors state that they "validated the KCC2–interactome by biochemically characterizing the interaction between KCC2 and the most abundant protein in the interactome, PACSIN1." This is an overstatement. The validation of the functional significance of PACSIN 1 is an important achievement, but it doesn't validate the significance of the entire interactome. It would be more accurate to state: "We validated the utility of the KCC2–interactome that we presented by verifying the direct biochemical interaction between pacsin1 and KCC2, and by demonstrating that pacsin1 participates in regulation of the level of expression of KCC2."

9) In the Discussion section, it is not clear to what the term "this detergent" refers. Do you mean C12E9 or CHAPS?

---

## [Author Response]

1) In the Materials and methods section, the authors must include more detail about the design and timing of the electrophysiology experiments. At what age were the neurons transfected? In each case, how long after the transfection were the recordings made? List explicitly the composition of the extracellular solution. For Figure 6 were the holding potentials set through the whole cell pipette? At what point in the IPSC was the amplitude measured? In the Results section discussing Figure 6 paragraph discussing each experiment should begin with the purpose of the experiment, carefully explain the design of the experiment, then end with a statement of what the results of the experiment mean. For panel 6C, fix the discrepancy between the units in the figure (pS), and the statement in subsection “PACSIN1 is a negative regulator of KCC2 expression and function in hippocampal neurons”, which lists the results as mV. Provide a separate paragraph explaining the rationale, execution, and conclusions from the gramicidin experiment. What were the "control cells" for this experiment? What were the chloride concentrations applied for control and gramicidin cells? Why is EGABA for control cells –10 mV; whereas in the shRNA control cells, the EGABA was –30 mV? An experiment must be performed in which the gramicidin perforated–patch recordings for the identification of the chloride reversal potential in the PACSIN1 shRNA KD neurons are compared with gramicidin perforated patch recordings in neurons expressing non–sense shRNA.

We have significantly expanded the Materials and methods and Results sections to include more detail about the electrophysiology experiments to ensure that the design and timing of the electrophysiology experiments is clear. Each Results section paragraph clearly states the purpose and design of the experiment, and ends with a statement interpreting the results. Specifically, we have included:

The age the neurons were transfected, the time after transfection when the recordings were made (Results section; Materials and methods section).

The composition of the extracellular recording solution is listed explicitly (Materials and Materials and methods section).

Explicitly stated that the holding potentials set through the whole cell pipette (for Figure 6; Results section, Materials and methods section).

Explicitly stated the point in the IPSC that the amplitude was measured (Materials and methods section).

For panel 6C, we have fixed the discrepancy between the units in the figure (pS), and the statement in subsection “PACSIN1 is a negative regulator of KCC2 expression and function in hippocampal neurons “(previous version), which accidently listed the results as mV (Results section).

What the "control cells" were for the gramicidin experiment (scrambled shRNA; Results section).

We now include a separate paragraph on the Gramicidin results (subsection “PACSIN1 is a negative regulator of KCC2 expression and function in hippocampal neurons”), where we have removed the comparison in the change in ^–^Cl^–^ between whole–cell and gramicidin–perforated patch clamp recordings, and now simply report E_GABA_ for gramicidin–perforated patch clamp experiments. We have also explained why both whole–cell and gramicidin–perforated patch clamp recordings were performed, and how they should be interpreted.

2) For Figure 6, provide information in the Materials and methods section about how the "fluorescence intensity was calculated. Do these numbers reflect total intensity over the whole field at each wavelength? Or were the images segmented in some way?

We have added the following statement to the Materials and methods section to clarify how we obtained the fluorescence intensity measurements: “Using ImageJ, 4 bisecting lines were drawn across the center of the cell. The peak values of each line (2 values/line) were used to calculate peak fluorescent intensity of KCC2 at the membrane.” We have also added a Supplemental Figure (Figure 6—figure supplement 1, which provides an example illustration of this calculation in ImageJ). Fluorescence intensities are plotted in arbitrary units (a.u.)

3) Although no significant change was observed in Figure 6, that there was a trend to larger GABA chloride conductance's in the pacsin KD neurons raises the question whether the KD of PACSIN1 may potentially have influenced the expression of other chloride channels. More importantly, the chloride reversal potential depends not only on the chloride conductance but also on the membrane potential. The authors should provide information about the resting potentials in the control and KD neurons.

We have provided information on the resting potentials in the control and KD neurons. Specifically, we have added Figure 6, which demonstrates that there was no significant difference in the resting membrane potential.

With respect to the proteomics experiments, several clarifications are needed.1) In the Materials and methods section, it is stated that the experiments providing data shown in Figure 2 and Figure 3 were performed only once. These experiments should be repeated at least one more time.

We have repeated these experiments to obtain an independent biological replicate; in total we now have three replicates; the pilot from Figure 2 = 1), the original replicate in Figure 3 = 2), and this new replicate added to Figure 3 = 3). A description of this repetition/replicate can be found in the Materials and methods section. Modifications to the interactome data (based on this additional replicate) are in Figure 3. Based on this new replicate data we added a fourth category (PLATINUM) to our categorization of KCC2 interactors; this additional category contains proteins originally in the GOLD category, that were replicated in 2 out of the 3 replicates.

2) In subsection “Determining Affinity Purification (AP) conditions to extract native–KCC2”, the chemical names of the three detergents (C12E9, CHAPS, and DDM) should be given at this first use of their abbreviated names. Similarly, in the Materials and methods section, the sources of each of the listed detergents should be given (i.e. where they were purchased).

We have provided the chemical names of all detergents (at first use), and have added the source of these detergents to the Materials and methods section.

3) Early in the manuscript, the relationship between the terms KCC2a, KCC2b, and SLC12A5 should be clearly stated. Is SLC12A5 the name of the gene that encodes KCC2a and b? If so, that term shouldn't be listed under the heading of Protein ID in Table 1. Should the heading instead by Gene name? It is a confusing practice to conflate the name given to a gene with the name of the protein. It leads to poor communication across research fields.

We have explained the relationship between the KCC2 isoforms and the gene SLC12a5 at the end of the first paragraph of the Introduction. We have also changed the heading in Table 1 and Figure 3 from ‘Protein ID’ to ‘Protein name’.

4) Similarly, in subsection “Members of KCC2 interactome are highly represented at excitatory synapses”, the meaning of SLC12A5 mRNA should be clarified. Does this term include mRNA encoding both the KCC2 isoforms? (Note that the word "members" is duplicated.)

The RNA seq data from the Allan Brain Atlas does not distinguish between isoforms KCC2a and KCC2b. We have added the following sentence to clarify (subsection “Members of KCC2 interactome are highly represented at excitatory synapses”): The RNAseq data does not distinguish between isoforms KCC2a and KCC2b, which equally represented in the neonatal brain, while KCC2b is the predominant isoform in the adult brain.

5) The legend for Figure 4 is incorrect and appears incomplete. The circles in 4A are red not pink. The meaning of the blue circles in b should be stated. The legend states that the thickness of the edge denotes the number of spectral enrichment, however, it would be clearer to say the thickness of the "radial line" or some other term since the lines in question don't appear as an edge. Upon close examination, one can see that the radial lines and the very light grey lines in 4A and 4B are identical. However, in 4A the grey lines appear to pass over the radial lines; whereas, in 4B they are behind the radial lines. It would be better to make the radial and background lines in the two figures identical so that the only differences are the red and blue circles denoting association with excitatory or inhibitory synapses. The explanation of the depiction of the mapping in these two figures should be elaborated in the figure legend. It is too terse, and makes the figure difficult to understand. Similarly, the listing of numbers of proteins in various groups that appears underneath the two circles in 4A and 4B is unexplained and seems out of place. It would be better to list those numbers in the figure legend, in association with a more complete description of the circular depictions.

We have corrected the colors in Figure 4, and we have indicated the meaning of the red, blue and pink circles under the figures in parentheses. We have changed the term “edge” to “radial lines” in the figure legends. We have also explained the meaning of black lines in the foreground and grey radial lines in the background in the figure legends. Additionally, we have made the foreground and background radial lines identical in 4A and 4B. We have explained this elaborately in the Materials and methods section titled ‘Integrated PPI network analyses and data representation’.

6) The legend of Figure 5, or the description of Figure 5 subsection “PACSIN1 is a novel native–KCC2 binding partner” should give the amino acid residue boundaries of the F–bar, SH3, and VAR regions in pacsin.

We have added the amino acid residue boundaries for the PACSIN1 F–bar, SH3, and VAR regions to the Results section.

7) The left side of Figure 5 appears to be mislabeled with IgG that should be IgY according to the legend.

Corrected.

8) In the Discussion section, the authors state that they "validated the KCC2–interactome by biochemically characterizing the interaction between KCC2 and the most abundant protein in the interactome, PACSIN1." This is an overstatement. The validation of the functional significance of PACSIN 1 is an important achievement, but it doesn't validate the significance of the entire interactome. It would be more accurate to state: "We validated the utility of the KCC2–interactome that we presented by verifying the direct biochemical interaction between pacsin1 and KCC2, and by demonstrating that pacsin1 participates in regulation of the level of expression of KCC2."

We agree with the reviewers that we overstated this significance of the validation. We have edited the statement as suggested. We now state (subsection “PACSIN1 is a negative regulator of KCC2 expression and function in hippocampal neurons”): “We validated the utility of the KCC2–interactome that we presented by verifying the direct biochemical interaction between PACSIN1 and KCC2, and by demonstrating that PACSIN1 participates in regulation of the level of expression of KCC2.”

9) In the Discussion, it is not clear to what the term "this detergent" refers. Do you mean C12E9 or CHAPS?

We are referring to CHAPS. This is now explicitly stated.